# PolyFlow: Safe and Efficient Polytope-Constrained Flow Matching with Constraint Embedding and Projection-free Update

**Jianming Ma** [1 2 *]  **Qiyue Yang** [1 2 *]  **Yang Zhang** [1]  **Liyun Yan** [1 2]  **Zhanxiang Cao** [1 2]  **Yazhou Zhang** [1]  **Yue Gao** [1 2]

## Abstract

While flow-based generative models have demonstrated strong performance across a wide range of domains, deploying them in safety-critical physical systems remains challenging due to strict constraint requirements. Existing approaches typically enforce safety through post-hoc corrections, which incur substantial computational overhead and may distort the learned distribution. We propose **PolyFlow**, a polytope-constrained flow matching framework that embeds constraints directly into the model and flow dynamics. PolyFlow introduces a discrete-time flow formulation and a projection-free architecture, which eliminate the discretization error and guarantee strict satisfaction of arbitrary polyhedral constraints, without the need for expensive iterative solvers. Experimental results show that PolyFlow achieves zero constraint violation while maintaining high distributional fidelity across a range of planning and control tasks. Compared to state-of-the-art constrained generation baselines, PolyFlow significantly reduces inference latency and demonstrates a favorable trade-off between safety, efficiency, and generative quality.

## 1. Introduction

Flow-based generative models have demonstrated remarkable performance and flexibility across a wide range of fields, ranging from high-fidelity image generation (Lipman et al., 2023) and video synthesis (Jin et al., 2025) to complex decision-making tasks (Zheng et al., 2023). These advances have motivated a growing transition toward deploying these powerful models in domains deeply rooted in the physical world, such as scientific discovery (Chen & Lipman, 2024), trajectory planning (Tan et al., 2025), and robotic control (Black et al., 2024).

Unlike traditional unconstrained generation tasks, physical applications are confronted with a fundamental challenge: *safety* and *constraint satisfaction*. In robot planning, for example, the environment is governed by strict geometric and dynamic constraints, including joint limits, obstacle avoidance (Li et al., 2025a), and actuation bounds (Orsolino et al., 2018). While an out-of-distribution pixel in image generation merely results in a visual artifact, a constraint violation in physical system can lead to hardware damage or catastrophic failure. This disparity highlights the need for generative models with explicit safety and constraint awareness.

Unfortunately, standard flow matching lacks inherent satisfaction for such safety (Yang et al., 2025). Even when both source and target distributions lie within the feasible region, standard flow matching often yields unsafe trajectories. Such violations stem from two primary sources: (1) **Fitting Error**: The parameterized neural model inevitably deviates from the ground-truth target vector field due to limited capacity or optimization difficulties, which may cause the flow to drift into unsafe regions. (2) **Numerical Integration Error**: Even if the learned ODE path is theoretically safe, the discretization required for sampling introduces numerical errors. Large integration steps—desirable for fast inference—potentially push the state across safety boundaries.

This challenge is further amplified in real-world applications where the training data itself may be unconstrained, yet we seek to enforce constraints during generation (Christopher et al., 2024). In such cases, naively matching the target distribution can result in conflicts with safety constraints.

Existing constrained generative methods typically rely on *post-hoc correction*, such as projection (Dai et al., 2025; Christopher et al., 2024) or reflection (Xie et al., 2024) operations applied during sampling. Although these methods can mitigate constraint violations to some extent, they suffer from notable limitations. Performing projections onto complex polytopes or manifolds is computationally expensive, creating a bottleneck for real-time inference. Furthermore,

---

*Equal contribution [1]Shanghai Jiao Tong University, Shanghai, China [2]Shanghai Innovation Institute, Shanghai, China. Correspondence to: Yue Gao <yuegao@sjtu.edu.cn>.

*Proceedings of the 43rd International Conference on Machine Learning*, Seoul, South Korea. PMLR 306, 2026. Copyright 2026 by the author(s).

such external interventions act as interference to the learned flow dynamics, which may distort the distribution matching and trap the sampling process in local optima.

To address these challenges, we propose a novel philosophy: *embedding constraints directly into the flow definition and model architecture*, rather than treating them as an afterthought. Following this principle, we present **PolyFlow**, a training paradigm designed for the strict satisfaction of arbitrary time-varying polyhedral constraints. First, to theoretically eliminate the risk of numerical integration error, PolyFlow reformulates the problem from continuous-time ODEs to a **discrete-time flow** framework. Second, to ensure the parameterized model inherently respects boundaries, we develop a **projection-free architecture** inspired by the Frank-Wolfe algorithm and ray-shooting techniques. This design guarantees that every update step resides within the feasible set by construction, eliminating the need for costly projection or optimization solvers during inference. Our main contributions are summarized as follows:

- **Discrete-Time Flow Formulation:** We reformulate flow matching in a discrete-time setting and provide theoretical analysis. We prove that the interior safety of conditional flows guarantees the safety of the marginal flow, thereby eliminating numerical integration errors as a source of constraint violation.

- **Projection-Free Architecture:** We propose PolyFlow, a novel parameterization that combines ray-shooting with learnable gating factors. This architecture ensures strict satisfaction of convex constraints by design while avoiding the computational overhead of iterative projection or optimization solvers.

- **Experimental Validation:** We demonstrate that PolyFlow achieves zero constraint violation and superior distribution matching quality compared to baselines across various constrained generation tasks, while significantly reducing inference latency.

## 2. Related Works

In safe-critical fields, generative models must not only capture high-dimensional distributions but also strictly adhere to physical laws and geometric boundaries. As summarized in Table 1, existing approaches can be broadly categorized into two paradigms:

**Post-hoc Correction** These approaches enforce safety constraints by adjusting trajectories during the sampling phase. *Projection-based* methods (Christopher et al., 2024; Luan et al., 2026; Jung et al., 2025) frame each sampling step as a constrained optimization problem, utilizing projection operators to map unsafe states back to the feasible region, which offers good scalability to complex constraints.

*Table 1.* Qualitative comparison of constrained generation frameworks. **Constraint Generalization** refers to the ability to handle diverse constraint types.

| Methods | Constraint Generalization | Inference Speed |
|---|---|---|
| Projection-based | ● ● ● | ● ○ ○ |
| Reflection-based | ● ○ ○ | ● ● ● |
| CBF-based | ● ● ○ | ● ○ ○ |
| Riemannian flow | ● ○ ○ | ● ● ● |
| Gauge map / Mirror map | ● ● ○ | ● ● ● |
| **PolyFlow (Ours)** | ● ● ● | ● ● ● |

Alternatively, *CBF-based* methods (Xiao et al., 2023; Dai et al., 2025; Cosner et al., 2024; Liu et al., 2025) introduce auxiliary control inputs via Control Barrier Functions (CBFs) to ensure forward invariance of the safety set. However, designing valid CBFs for general constraints remains a challenging task.

Despite their different formulations, both families of methods share two fundamental limitations. First, they generally require solving Quadratic Programming (QP) problems iteratively, which substantially increases inference latency and computational cost. Second, strong post-hoc safety corrections act as external disruptions to the learned dynamics, potentially trapping the generation process in local minima or distorting the target distribution.

**Intrinsic Constraint Embedding** Instead of correcting violations after they occur, these methods embed constraints directly into the flow dynamics. *Reflection-based* models (Lou & Ermon, 2023; Xie et al., 2024; Fishman et al., 2023) incorporate reflection terms into SDEs or ODEs to confine the process within boundaries, but deriving reflection terms for complex, non-smooth geometries is mathematically difficult. *Riemannian flow* approaches (Davis et al., 2024; Cheng et al., 2024) define flow directly on constrained manifolds, yet their applicability is largely limited to simple geometries due to the difficulty of constructing suitable Riemannian metrics for arbitrary constraints. More recently, *Gauge map* and *Mirror map* methods (Li et al., 2025b; Guan et al., 2025; Feng et al., 2024) transform constrained domains into simpler unconstrained spaces via bijective mappings, where standard flow or diffusion models can be trained, and then map samples back to the original domain. However, these approaches rely on carefully designed mappings or distance functions and may introduce additional approximation errors or hyperparameter sensitivity.

A central challenge in constrained flow matching is the trade-off between *distribution matching fidelity* and *strict constraint satisfaction*. Post-hoc strategies decouple constraint enforcement from training, often leading to conflicts between the learned flow and safety filter. Conversely, intrinsic methods integrate constraints seamlessly but suffer from limited scalability and generalization to complex ge-

ometries. Consequently, it remains an open problem to develop a general, computationally-efficient framework that embeds hard constraints into the flow architecture without compromising distribution quality.

## 3. Background

Standard Flow Matching (FM) models a *continuous* probability path $p_t(\cdot)$ that transforms a simple prior distribution $p_0$ to a complex target distribution $p_1$. This transformation is governed by a time-dependent vector field $u_t : [0, 1] \times \mathbb{R}^d \to \mathbb{R}^d$, which generates the flow $\psi_t$ via the ordinary differential equation (ODE):

$$\frac{d}{dt}\psi_t(x) = u_t(\psi_t(x)), \quad \psi_0(x) = x, \tag{1}$$

where $\psi_1(x)$ pushes samples from $p_0$ to $p_1$. The goal is to learn a parameterized vector field $u_\theta$ that approximates the ground-truth marginal vector field $u_t$ by minimizing the objective:

$$\mathcal{L}_{\text{FM}}(\theta) = \mathbb{E}_{t,x_t} \left[ \|u_\theta(x_t, t) - u_t(x_t)\|^2 \right], \tag{2}$$

where $t \sim \mathcal{U}[0, 1]$, $x_t \sim p_t$. However, the marginal vector field $u_t$ is generally intractable. To address this, Conditional Flow Matching (CFM) (Lipman et al., 2023) proposes working with conditional vector field $u_t(x|z)$ conditioned on a latent variable $z$, typically defined as a data pair $z = (x_0, x_1)$. Crucially, CFM proves that regressing $u_\theta$ towards the conditional vector field yields the same gradients as the intractable FM objective:

$$\mathcal{L}_{\text{CFM}}(\theta) = \mathbb{E}_{t,z,x_t} \left[ \|u_\theta(x_t, t) - u_t(x_t|z)\|^2 \right], \tag{3}$$
$$\nabla_\theta \mathcal{L}_{CFM} = \nabla_\theta \mathcal{L}_{FM} \tag{4}$$

where $t \sim \mathcal{U}[0, 1]$, $z \sim q_z$, $x_t \sim p_t(x_t|z)$.

In practice, a linear probability path (Optimal Transport path) is often adopted, defined as $x_t = (1 - t)x_0 + tx_1$. This choice yields a closed-form conditional vector field $u_t(x_t|x_0, x_1) = x_1 - x_0$. Consequently, the loss function simplifies to a simple regression problem:

$$\mathcal{L}_{\text{CFM}}(\theta) = \mathbb{E}_{t,x_0 \sim p_0, x_1 \sim p_1} \left[ \|u_\theta(x_t, t) - (x_1 - x_0)\|^2 \right]. \tag{5}$$

## 4. Method

While continuous flow matching provides an effective paradigm for unconstrained generative modeling, enforcing strict safety constraints during ODE integration is inherently difficult, as numerical integration errors accumulate and can violate feasibility boundaries. In this section, we introduce a discrete-time framework and provide rigorous theoretical analysis to justify discrete-time flow matching. Based on the

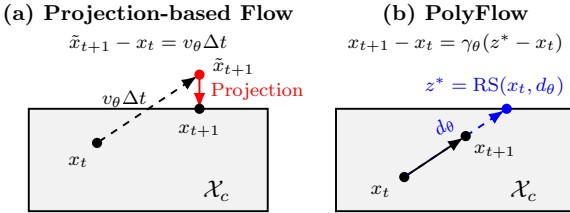

*Figure 1.* Illustration of PolyFlow update.

discrete-time flow, we propose **PolyFlow**, a projection-free architecture designed to ensure strict safety satisfaction in constrained generation tasks.

### 4.1. Discrete-Time Flow and Safety

Let $\mathcal{T} = \{0, 1, \ldots, T - 1\}$ denote discrete time steps. We consider a dynamical system governed by a sequence of vector fields $\{u_t\}_{t \in \mathcal{T}}$, where $u_t : \mathbb{R}^d \to \mathbb{R}^d$.

**Definition 4.1** (Discrete-Time Flow). The discrete-time flow $\psi : \{0, \ldots, T\} \times \mathbb{R}^d \to \mathbb{R}^d$ is defined by the following difference equation:

$$\begin{cases} \psi_{t+1}(x) = \psi_t(x) + u_t(\psi_t(x)) \\ \psi_0(x) = x \end{cases} . \tag{6}$$

Given an initial distribution $x \sim p_0$, this flow induces a probability path denoted by $p_t$, where $\psi_t(x) \sim p_t$ for $t = 0, \ldots, T$.

We focus on generation within a safe region. Let $\mathcal{X}_c \subset \mathbb{R}^d$ be a compact and convex set representing the feasible space.

**Definition 4.2** (Interior Safe Discrete-Time Flow). A discrete-time flow generated by $\{u_t\}_{t \in \mathcal{T}}$ is *interior safe* with respect to $\mathcal{X}_c$, if for any initial state $x_0 \sim p_0, x_0 \in \mathcal{X}_c$, the resulting trajectory $\psi_t(x_0)$ remains in the feasible set:

$$\psi_t(x_0) \in \mathcal{X}_c, \quad \forall t \in \{0, \ldots, T\}. \tag{7}$$

This implies recursive safety: if $x_t \in \mathcal{X}_c$, then $x_{t+1} = x_t + u_t(x_t) \in \mathcal{X}_c$.

### 4.2. Discrete-Time Flow Matching

To train the discrete-time flow, we adopt the CFM framework (Lipman et al., 2023) adapted for discrete-time dynamics. Let $z \sim p_Z(z)$ denote a sample of latent variables. We define a conditional vector field $u_t(x|z)$ that generates a conditional probability path $p_{t|Z}(x|z)$. The marginal distribution is as follows:

$$p_t(x) = \int p_{t|Z}(x|z)p_Z(z)dz, \quad \forall t \in \{0, \ldots, T\}, \tag{8}$$

Similarly to CFM, we further define the marginal expectation field $\tilde{u}_t(x)$:

$$\tilde{u}_t(x) \triangleq \mathbb{E}_{z \sim p_{Z|t}(\cdot|x)}[u_t(x|z)], \quad \forall t \in \mathcal{T}. \tag{9}$$

A parameterized vector field $u_\theta$ is trained to approximate this field via the CFM objective:

$$\mathcal{L}_{\text{CFM}} = \mathbb{E}_{z,t}\mathbb{E}_{x_t \sim p_{t|Z}(\cdot|z)}\left[\|u_\theta(x_t, t) - u_t(x_t|z)\|^2\right]. \quad (10)$$

*Remark* 4.3. Following the theoretical framework of CFM (Lipman et al., 2023), the parameterized vector field $u_\theta$ trained via Equation (10) approximates the marginal expectation field $\tilde{u}_t$ rather than the true marginal vector field. In the continuous-time setting, these two fields are equivalent and generate the same marginal path. However, in the discrete-time setting, this equivalence no longer holds: $\tilde{u}_t$ does not, in general, exactly generate the true marginal path $p_t$, introducing a discretization error that must be carefully bounded.

Let $\tilde{p}_t$ denotes the approximate probability path generated by iterating $\tilde{u}_t$, we quantify the discrepancy between this two probability paths using the 2-Wasserstein distance.

**Theorem 4.4** (Discretization Error Bound). *Assume that the marginal expectation field $\tilde{u}_t(x)$ is L-Lipschitz continuous. Let $W_2(p_t, \tilde{p}_t)$ denote the 2-Wasserstein distance between $p_t$ and $\tilde{p}_t$. Then at step $t + 1$:*

$$W_2(p_{t+1}, \tilde{p}_{t+1}) \le (1 + L)W_2(p_t, \tilde{p}_t) + \mathcal{E}_{match}(t).$$

*where $\mathcal{E}_{match}(t) = \sqrt{\mathbb{E}_z\mathbb{E}_{x_t \sim p_{t|Z}}[\|\tilde{u}_t(x_t) - u_t(x_t|z)\|^2]}$ represents the matching error.*

Detailed proofs for all theorems are provided in Appendix B.

Theorem 4.4 reveals that the generation error accumulates recursively and is controlled by the matching error $\mathcal{E}_{\text{match}}(t)$. When the CFM objective in Equation (10) is minimized, $\mathcal{E}_{\text{match}}(t)$ is small, and the approximation error remains bounded over all $T$ steps. This justifies the use of $\tilde{u}_t$ as a tractable proxy for discrete generation.

Although discretization inevitably introduces minor deviations in the marginal distribution, the discrete-time flow framework theoretically circumvents the numerical integration errors inherent to continuous ODE sampling. This structural advantage enables rigorous safety guarantees directly within the discrete formulation.

**Theorem 4.5** (Safety Preservation). *Let $\mathcal{X}_c$ be a convex set. If the conditional flow is interior safe (Definition 4.2), then the marginal expectation field $\tilde{u}_t(x)$ is also interior safe.*

Theorem 4.5 guarantees that aggregating safe conditional flows results in a safe marginal flow. To satisfy the preconditions of this theorem, we construct the conditional flow using linear interpolation between endpoints $x_0, x_1 \in \mathcal{X}_c$. The conditional field is defined as $u_t(x|z) = (x_1 - x_0)/T$. Due to the convexity of $\mathcal{X}_c$, any point on the segment connecting $x_0$ and $x_1$ remains within $\mathcal{X}_c$. Consequently, the

---

**Algorithm 1** PolyFlow Training

1: **Input:** Source and target dist. $p, q$, Safety Polytope $\mathcal{X}_c$.
2: **Initialize:** Direction network $d_\theta$, Gating network $\gamma_\theta$.
3: **while** not converged **do**
4:     Sample batches $x_0 \sim p$, $x_1 \sim q$.
5:     Sample time steps $t \sim \mathcal{U}(\{0, \ldots, T-1\})$.
6:     $x_t \leftarrow (1 - \frac{t}{T})x_0 + \frac{t}{T}x_1$     ▷ *Interpolant state*
7:     $u_{gt} \leftarrow \frac{1}{T}(x_1 - x_0)$     ▷ *Target discrete update*
8:     $d \leftarrow d_\theta(x_t, t, \mathcal{X}_c)$     ▷ *Predict raw direction*
9:     $z^* \leftarrow \text{RS}(x_t, d, \mathcal{X}_x)$     ▷ *Ray Shooting to $\partial\mathcal{X}_c$*
10:    $\gamma \leftarrow \text{Sigmoid}(\gamma_\theta(x_t, t, \mathcal{X}_c))$    ▷ *Predict step size*
11:    $u_\theta \leftarrow \gamma \cdot (z^* - x_t)$    ▷ *Construct safe update*
12:    $\mathcal{L} \leftarrow \|u_\theta - u_{gt}\|^2$
13:    Update $\theta \leftarrow \theta - \eta\nabla_\theta\mathcal{L}$
14: **end while**

---

conditional flow is interior safe, and by Theorem 4.5, the marginal expectation field $\tilde{u}_t$ also preserves these safety constraints.

### 4.3. PolyFlow: Safe Discrete-Time Flow Matching

Although Theorem 4.5 guarantees safety for the marginal field $\tilde{u}_t$, the parameterized network $u_\theta$ is only an approximation of the field. To enforce strict safety, we reformulate the CFM training objective as a constrained optimization problem:

$$\min_\theta \mathcal{L}_{CFM} = \mathbb{E}_z\mathbb{E}_{x_t \sim p_{t|Z}(\cdot|z)}\left[\|u_\theta(x_t, t) - u_t(x_t|z)\|^2\right]$$
$$\text{s.t.} \quad x_t + u_\theta(x_t, t) \in \mathcal{X}_c, \quad \forall x_t \in \mathcal{X}_c, \forall t \in \mathcal{T} \quad (11)$$

Many existing approaches enforce constraints by introducing a projection operator during sampling, i.e., $x_{t+1} = \text{Proj}_{\mathcal{X}_c}(x_t + u_\theta(x_t, t))$. Such post-hoc projections incur additional computational cost and may interfere with the learned flow dynamics. In contrast, we adopt a *projection-free* strategy that embeds geometric constraints directly into the architecture of the parameterized flow model.

**PolyFlow** gets inspiration from the Frank-Wolfe algorithm (see Appendix A), which addresses constrained optimization problems by moving towards vertices of the feasible set rather than projecting gradients.

The core insight is to parameterize the update vector $u_\theta$ as a scaled direction pointing towards a boundary element of the safety polytope. As a direct Frank-Wolfe analogy, one may consider the following formulation:

$$u_\theta(x_t, t) = \gamma_\theta(x_t, t) \cdot (\text{Selector}_\theta(\mathcal{V}(\mathcal{X}_c)) - x_t), \quad (12)$$

where $\mathcal{V}(\mathcal{X}_c)$ is the set of vertices of the safety polytope $\mathcal{X}_c$, $\text{Selector}_\theta$ denotes a module that selects a target vertex from this set, and $\gamma_\theta \in [0, 1]$ represents a learned adaptive

step size. By convexity, if $x_t \in \mathcal{X}_c$, the update guarantees $x_{t+1} \in \mathcal{X}_c$.

However, a direct implementation of Eq. (12) faces several practical challenges. First, the discrete nature of the vertex selection makes the architecture non-differentiable. Second, restricting update directions solely towards vertices—as in standard Frank-Wolfe—often results in "zig-zagging" behavior (Lacoste-Julien & Jaggi, 2015), leading to non-smooth trajectories.

To overcome these limitations, we decouple the update into two components: a freely-learned direction and a bounded step magnitude. Instead of constraining the update direction to point exclusively towards vertices, we allow the model to learn a continuous direction field and transfer it onto the safety set boundary using a *Ray Shooting* operation. Formally, the update rule is reformulated as:

$$u_\theta(x_t, t) = \gamma_\theta(x_t, t) \cdot (\mathrm{RS}(x_t, d_\theta(x_t, t)) - x_t), \quad (13)$$

where:

1) $d_\theta(x_t, t) : \mathbb{R}^d \times \mathbb{R} \to \mathbb{R}^d$ is a neural network predicting an unconstrained direction vector (analogous to the unconstrained gradient/score).

2) $\mathrm{RS}(x_t, d_\theta) : \mathcal{X}_c \times \mathbb{R}^d \to \partial\mathcal{X}_c$ is the **Ray Shooting** operator (see Appendix C). It computes the intersection point between the boundary $\partial\mathcal{X}_c$ and a ray originating at $x$ in the direction of $d$.

3) $\gamma_\theta(x_t, t) \in [0, 1]$ is a learned scalar gating factor (via a sigmoid output) that controls the step length along the line segment connecting the current state $x_t$ and the boundary intersection.

This parameterization offers distinct advantages. First, computing the ray-boundary intersection is differentiable with respect to both the origin $x_t$ and the direction $d_\theta$. Second, predicting a continuous direction $d_\theta$ allows the model to target any point on the polytope boundary (including faces and edges), substantially increasing the degrees of freedom. This flexibility enables the flow to generate smooth and efficient trajectories and avoid oscillatory behavior. Consequently, by combining projection-free philosophy with differentiable ray shooting, PolyFlow achieves strict safety satisfaction by construction while maintaining training tractability and trajectory smoothness. Pseudo codes for the training and sampling process are presented in Algorithm 1 and 2.

### 4.4. Practical Implementation

**Initial Sampling**    According to Definition 4.2 and Theorem 4.5, strict constraint satisfaction requires that the support of the initial distribution lies entirely in the feasible region, $\mathrm{supp}(p) \subset \mathcal{X}_c$. Therefore, we construct the initial distribution by uniformly sampling within the Chebyshev

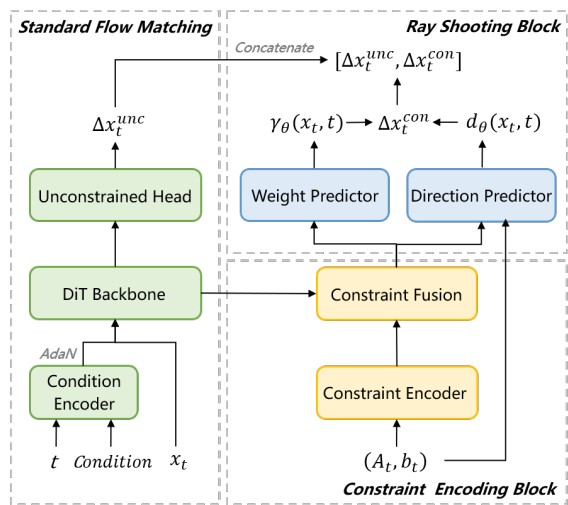

*Figure 2.* Overall framework of PolyFlow.

ball of the convex polytope $\mathcal{X}_c$. Calculating the Chebyshev ball is formulated as a Linear Programming (LP) problem, which is computationally cheaper than the QP projection used in projection-based methods. Furthermore, for tasks with static constraints, this LP problem needs to be solved only once during the pre-computation phase, resulting in negligible additional computational overhead during online sampling.

**Model Structure**    Our approach integrates seamlessly into existing flow model architectures. Figure 2 illustrates the overall framework of PolyFlow. Since constraints may apply to only a subset of dimensions of $x$, we partition $x$ into an unconstrained component $x^{unc}$ and a constrained component $x^{con}$, which are processed by separate output heads. For the constrained component, we augment the standard DiT (Peebles & Xie, 2023) backbone with two key modules: a *Constraint Encoding Block* and a *Ray Shooting Block*. The Constraint Encoding Block takes the set of linear inequality constraints $\mathcal{C} = \{(A_i, b_i) \mid A_i \in \mathbb{R}^n, b_i \in \mathbb{R}\}_{i=1}^m$ (i.e., $Ax \leq b$) as input. To handle the permutation-invariant nature of the constraint set, we employ Transformer blocks without positional encoding (Lee et al., 2019). The resulting constraint embeddings are fused with the latent representation of the DiT backbone via a cross-attention mechanism. The Ray Shooting Block subsequently takes the fused representation as input and predicts two quantities: a weight $\gamma_\theta(x_t, t)$ and a direction $d_\theta(x_t, t)$, which together parameterize the constrained update step $\Delta x_t^{con}$.

## 5. Experiments

We design experiments to address the following questions:

**Q1**: Can PolyFlow ensure strict constraint satisfaction across diverse tasks?

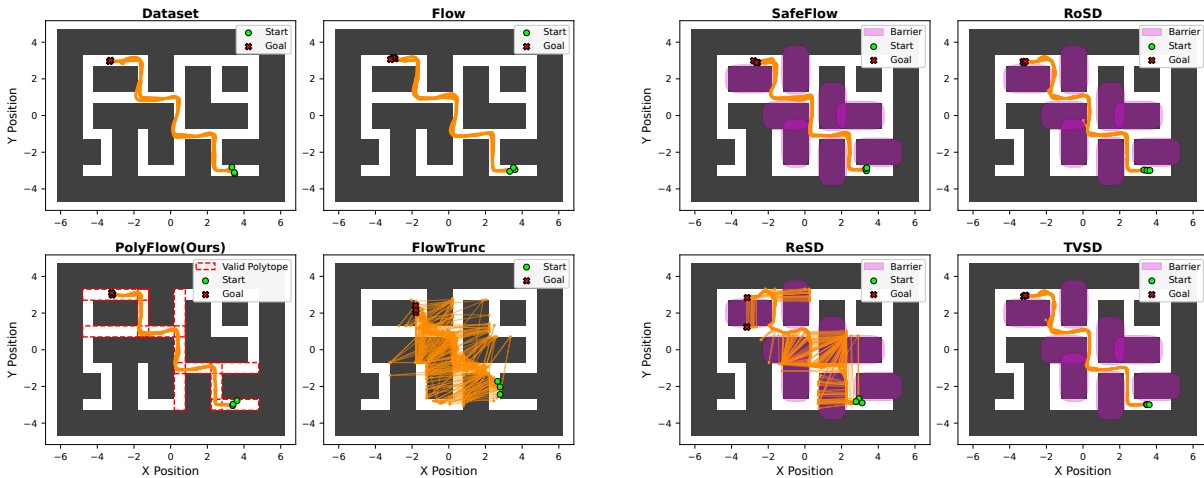

*Figure 3.* Generated trajectories of different methods in Maze task.

**Q2**: Does PolyFlow achieve a superior trade-off between task performance and inference efficiency compared to strong constrained baselines?

**Q3**: Can PolyFlow effectively handle *dynamic* and *complex* constraint formulations?

### 5.1. Experimental Setup

**Baselines** To evaluate the performance of our method, we compare it against several representative constrained generation approaches: (1) **SafeFlow** (Dai et al., 2025), a CBF-based method that incorporates forward-invariance control signals into the flow matching vector field to guide generation; (2) **SafeDiffuser** variants (**RoSD/ReSD/TVSD**) (Xiao et al., 2023), which integrate CBFs into diffusion models using distinct guidance strategies; (3) **FlowTrunc**, a projection-based baseline that enforces constraints by truncating violations during generation; and (4) **GaugeFlow** (Li et al., 2025b), which maps the feasible region to a unit ball via gauge maps and utilizes reflection for constraint satisfaction.

**Tasks Description** The proposed method is validated across three representative constrained robotic planning and control benchmarks, ranging from low-dimensional navigation to high-dimensional quadrupedal locomotion.

*1) 2D Maze Navigation.* This serves as a standard benchmark for testing constraint satisfaction in trajectory generation. The agent must generate strictly collision-free paths from a start to a goal. Although CBF-based baselines approximate the safe set using a series of hyper-ellipsoids, PolyFlow decomposes the free space into a sequence of convex safe corridors (see Appendix F), as illustrated in Figure 3.

*2) Gym Locomotion.* We select five constrained closed-

loop control tasks of varying complexity: *Hopper-Simple*, *Hopper-Complex*, *Walker2d-Simple*, *Walker2d-Complex*, and *HalfCheetah*. For the Hopper and Walker2d tasks, constraints are imposed on the state space, whereas for HalfCheetah, constraints are applied directly to the action space. We define both *Simple* and *Complex* constraint formulations to test model robustness. (see Appendix G)

*3) Quadruped Locomotion.* To assess PolyFlow's capability in handling dynamic and complex constraints (**Q3**), we apply it to the control of a Unitree Go2 quadruped robot. In this challenging scenario, the action space is restricted by complex, time-varying friction cone constraints to ensure non-slip contact with the ground. (see Appendix H)

**Evaluation Metrics** We employ a comprehensive set of metrics (details in Appendix E) categorized into three dimensions to rigorously evaluate PolyFlow and the baselines:

*1) Safety Satisfaction.* We use the **Safety Rate (R)** to measure the proportion of generated trajectories that strictly satisfy all imposed constraints.

*2) Generative Quality.* To assess the fidelity of the generated distribution relative to the expert distribution, we compute the **Maximum Mean Discrepancy (MMD)**, **2-Wasserstein Distance (W)**, and **KL Divergence (KL)**. Furthermore, we evaluate the smoothness of the generated trajectories, specifically **Curvature Smoothness (Cur)** and **Acceleration Smoothness (Acc)**.

*3) Inference Efficiency.* We assess the practical viability of each method by measuring the average total inference time per trajectory (**TotalTime**) and the average computational time required for a single sampling step (**StepTime**).

*Table 2.* Metric comparisons for 2D Maze task. **(a)** shows the performance using the default sampling steps for baselines. **(b)** restricts all methods to exactly $N = 10$ sampling steps to provide a strict equal-computation comparison. **Bold** indicates the best performance.

**(a) Default Sampling Steps** (Flow/FlowTrunc/SafeFlow: N=200; RoSD/ReSD/TVSD: N=256)

| Metric | PolyFlow-attn | PolyFlow-mlp | Flow | FlowTrunc | SafeFlow | RoSD | ReSD | TVSD |
|---|---|---|---|---|---|---|---|---|
| Safety($\uparrow$) | **1.0** | **1.0** | 0.765 | 0.0 | 0.965 | 0.98 | 0.34 | 0.945 |
| MMD($\downarrow$) | $6.20 \times 10^{-6}$ | $\mathbf{2.62 \times 10^{-6}}$ | $2.57 \times 10^{-5}$ | 0.202 | $5.77 \times 10^{-3}$ | $3.55 \times 10^{-4}$ | $4.25 \times 10^{-3}$ | $7.62 \times 10^{-4}$ |
| W($\downarrow$) | 0.041 | **0.033** | 0.046 | 2.361 | 0.321 | 0.129 | 0.665 | 0.166 |
| KL($\downarrow$) | 0.134 | **0.050** | 0.168 | 9.093 | 2.842 | 38.057 | 1.043 | 65.709 |
| Cur($\downarrow$) | 0.098 | 0.097 | **0.084** | 2.075 | 0.164 | 0.086 | 1.065 | **0.084** |
| Acc($\downarrow$) | $6.15 \times 10^{-3}$ | $6.37 \times 10^{-3}$ | $\mathbf{4.98 \times 10^{-3}}$ | 2.00 | $9.13 \times 10^{-3}$ | $5.57 \times 10^{-3}$ | 1.46 | $5.25 \times 10^{-3}$ |
| TotalTime($\downarrow$) | 2.11 | **0.58** | 3.65 | 3.69 | 153.50 | 10.30 | 73.46 | 57.52 |
| StepTime($\downarrow$) | 0.211 | 0.058 | **0.018** | **0.018** | 0.767 | 0.040 | 0.299 | 0.225 |

**(b) Equal Sampling Steps** ($N = 10$)

| Metric | PolyFlow-attn | PolyFlow-mlp | Flow | FlowTrunc | SafeFlow | RoSD | ReSD | TVSD |
|---|---|---|---|---|---|---|---|---|
| Safety($\uparrow$) | **1.0** | **1.0** | 0.845 | 0.0 | 0.530 | 0.990 | 0.145 | 0.0 |
| MMD($\downarrow$) | $6.20 \times 10^{-6}$ | $\mathbf{2.62 \times 10^{-6}}$ | $3.50 \times 10^{-5}$ | $2.71 \times 10^{-2}$ | $6.55 \times 10^{-3}$ | $1.76 \times 10^{-4}$ | $5.46 \times 10^{-2}$ | $9.01 \times 10^{-2}$ |
| W($\downarrow$) | 0.041 | **0.033** | 0.054 | 0.895 | 0.348 | 0.124 | 1.800 | 2.450 |
| KL($\downarrow$) | 0.134 | **0.050** | 0.266 | 7.860 | 3.140 | 50.200 | 2.720 | 3.740 |
| Cur($\downarrow$) | 0.098 | **0.097** | 0.193 | 1.951 | 1.084 | 0.150 | 1.548 | 2.094 |
| Acc($\downarrow$) | $\mathbf{6.15 \times 10^{-3}}$ | $6.37 \times 10^{-3}$ | $1.14 \times 10^{-2}$ | 0.875 | $9.11 \times 10^{-2}$ | $1.76 \times 10^{-2}$ | 2.288 | 4.223 |
| TotalTime($\downarrow$) | 2.11 | 0.58 | **0.324** | 0.363 | 7.558 | 1.120 | 8.926 | 2.558 |
| StepTime($\downarrow$) | 0.211 | 0.058 | **0.032** | 0.036 | 0.756 | 0.112 | 0.893 | 0.256 |

## 5.2. Safe Planning in 2D Maze

We evaluate the performance of PolyFlow against several baselines in the 2D Maze environment. Table 2 (a) presents the results using the default sampling steps for all baselines to demonstrate their peak performance, while Table 2 (b) provides a strict comparison under an identical low-step regime ($N = 10$). Furthermore, we investigate two architectural variants for PolyFlow's constraint encoding block: *PolyFlow-attn*, which utilizes self-attention for encoding and cross-attention for feature fusion, and *PolyFlow-mlp*, which employs a MLP with pooling for feature extraction and additive fusion.

**Safety Performance** As shown in Table 2, both PolyFlow variants achieve strict constraint satisfaction (100% safety rate) across all settings. In contrast, all baselines fail to guarantee absolute safety even with extensive sampling steps[cite: 191]. Moreover, in the equal-step comparison (Table 2b), the safety of post-hoc correction methods collapses significantly. For CBF-based approaches (SafeFlow, RoSD, ReSD, and TVSD), modeling complex mazes requires approximating obstacles with multiple super-ellipses. As the number of constraints increases, the safety controller often becomes infeasible, necessitating the use of slack variables that inherently compromise theoretical safety guarantees. PolyFlow circumvents these issues by embedding constraints directly into the architecture.

**Generational Quality** PolyFlow-mlp achieves state-of-the-art performance across all distributional metrics, notably outperforming even the unconstrained Flow matching baseline. In addition, it is observed in Figure 3 that methods such as ReSD and FlowTrunc suffer from severe oscillations, with trajectory points clustered near obstacle boundaries. This highlights a classic "local trap" issue in post-hoc correction methods: when external forcing becomes too aggressive, it disrupts the natural ODE dynamics and traps the generation process in local minima, yielding physically non-viable paths.

**Inference Efficiency** A key advantage of PolyFlow's projection-free architecture is its computational efficiency. According to Table 2(b), when fixed at 10 steps, the total inference time of PolyFlow-mlp (0.58s) is orders of magnitude faster than SafeFlow (7.558s) and nearly matches the unconstrained Flow (0.324s). Interestingly, the MLP-based encoder proves much more efficient than the Attention-based counterpart, reducing total time by approximately 72% while simultaneously improving trajectory quality. This demonstrates that for polyhedral constraints, a simplified encoding structure is sufficient to achieve superior performance with minimal overhead, making PolyFlow-mlp highly suitable for real-time safety-critical applications.

*Table 3.* **Generative metric comparison across five tasks**. The *Dismatching Score* is calculated as $\frac{1}{3}\left(\frac{\text{MMD}}{\text{MMD}_{\text{Flow}}} + \frac{\text{W}}{\text{W}_{\text{Flow}}} + \frac{\text{KL}}{\text{KL}_{\text{Flow}}}\right)$ relative to the Flow baseline (detailed in Table 8). The return is calculated by rolling out with 10 different seeds. We use PolyFlow-attn in these tasks.

| Task | Metric | Flow | PolyFlow (Ours) | SafeFlow | RoSD | GaugeFlow |
|---|---|---|---|---|---|---|
| **Hopper-Simple** | Dismatching Score ($\downarrow$) | – | **0.899** | 0.984 | 5.068 | 1.049 |
| | Total Time (s) ($\downarrow$) | 0.698 | **0.075** | 0.833 | 1.749 | 0.867 |
| | Safety Rate ($\uparrow$) | 0.755 | 1.000 | 1.000 | 1.000 | 1.000 |
| | Rollout Return ($\uparrow$) | $2450 \pm 878$ | $\mathbf{3187 \pm 753}$ | $2628 \pm 887$ | $961 \pm 613$ | $2473 \pm 951$ |
| **Hopper-Complex** | Dismatching Score ($\downarrow$) | – | **1.022** | 1.169 | 27.120 | 1.189 |
| | Total Time (s) ($\downarrow$) | 0.698 | **0.081** | 15.125 | 2.114 | 0.878 |
| | Safety Rate ($\uparrow$) | 0.005 | 1.000 | 1.000 | 1.000 | 1.000 |
| | Rollout Return ($\uparrow$) | $2450 \pm 878$ | $\mathbf{2949 \pm 838}$ | $2397 \pm 877$ | $937 \pm 539$ | $2851 \pm 944$ |
| **Walker2d-Simple** | Dismatching Score ($\downarrow$) | – | **0.892** | 1.000 | 1.240 | 1.041 |
| | Total Time (s) ($\downarrow$) | 0.403 | **0.081** | 0.598 | 2.043 | 0.649 |
| | Safety Rate ($\uparrow$) | 0.985 | 1.000 | 1.000 | 1.000 | 1.000 |
| | Rollout Return ($\uparrow$) | $5895 \pm 113$ | $\mathbf{6031 \pm 89}$ | $5936 \pm 96$ | $5816 \pm 224$ | $5974 \pm 72$ |
| **Walker2d-Complex** | Dismatching Score ($\downarrow$) | – | **0.891** | 0.999 | 1.248 | 1.028 |
| | Total Time (s) ($\downarrow$) | 0.348 | **0.079** | 4.530 | 1.339 | 0.435 |
| | Safety Rate ($\uparrow$) | 0.705 | 1.000 | 1.000 | 1.000 | 1.000 |
| | Rollout Return ($\uparrow$) | $5895 \pm 113$ | $\mathbf{5981 \pm 114}$ | $5961 \pm 95$ | $5716 \pm 577$ | $5783 \pm 618$ |
| **HalfCheetah** | Dismatching Score ($\downarrow$) | – | 5.256 | 3.177 | **3.099** | 4.334 |
| | Total Time (s) ($\downarrow$) | 0.358 | **0.089** | 10.071 | 2.334 | 0.502 |
| | Safety Rate ($\uparrow$) | 0.000 | 1.000 | 1.000 | 1.000 | 1.000 |
| | Rollout Return ($\uparrow$) | $724 \pm 375$ | $\mathbf{2977 \pm 785}$ | $2034 \pm 892$ | $2083 \pm 580$ | $1399 \pm 309$ |

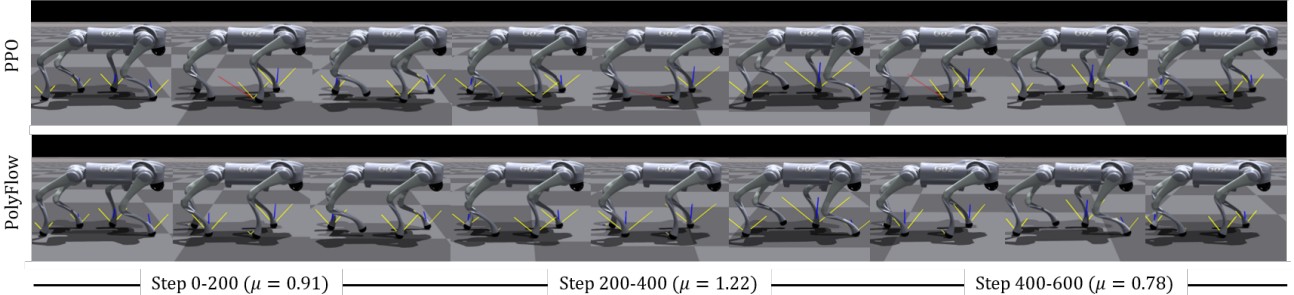

*Figure 4.* **Visualization of ground reaction forces in quadruped locomotion task.** Yellow lines indicate friction cone boundaries; blue and red lines correspond to safe and unsafe forces, respectively. Notably, PolyFlow strictly satisfies constraints throughout the rollout, in contrast to the expert policy which shows clear violations.

## 5.3. Safe Control for Gym Tasks

In the Gym locomotion experiments, we employ a receding horizon control framework. At each control step, the model predicts a trajectory over a future horizon, in which all generated states strictly satisfy the predefined safety constraints.

Table 3 presents the quantitative results for both trajectory matching and control rollouts. PolyFlow shows superior performance across the board, achieving strict constraint satisfaction during generation while attaining the highest rollout returns and the lowest inference latency in all tasks. Regarding generative fidelity, PolyFlow also yields the best fit to the expert distribution in four out of five tasks.

Another notable observation in Table 3 is that constrained methods (PolyFlow and SafeFlow) occasionally outperform the unconstrained Flow baseline in terms of task return. We hypothesize that this is because low-return trajectories in

the dataset often correlate with unsafe behaviors. Enforcing safety constraints therefore acts as "hard guidance", steering the model toward safer data modes that yield higher rewards.

To strictly evaluate this effect, we analyze constraint violations of the actual executed trajectories (rollouts), as shown in Table 9 in the Appendix. Since constraints are enforced on the generated states rather than directly on control actions, minor violations may occur during physical execution. However, constrained methods substantially reduce both the magnitude and duration of these violations. Among all methods, PolyFlow achieves the most favorable trade-off, securing the lowest constraint violation magnitude while maintaining the highest task return, as shown in Figure 5.

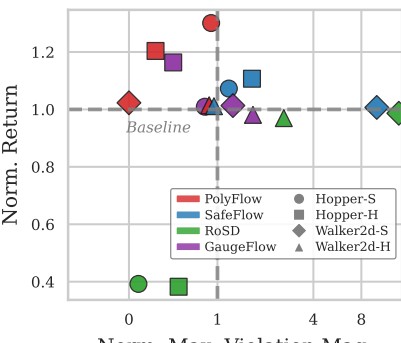

*Figure 5.* **Safety-return trade-off during rollouts.** This figure illustrates the Pareto frontier of different methods across *Max V. Mag.*(maximum violation magnitude ratio) and *Rollout Return*. Both metrics are normalized with respect to the Flow baseline. PolyFlow achieves a Pareto frontier positioned closer to the top-left compared to baselines, which indicates better safety-performance trade-off. (**Note**: For Hopper and Walker2d, constraints are applied to the states of generated trajectories instead of the executed actions. Consequently, constraint violations may occur during rollouts yet the imposed constraints provide effective guidance.)

### 5.4. Safe control for Quadrupedal Locomotion

We further evaluate the performance of PolyFlow under highly dynamic and time-varying safety constraints. As discussed in Appendix H, mapping the friction pyramid to action space introduces state-dependent variations (illustrated in Figure 11), making the feasible polytope naturally time-varying as the robot moves. In addition, we dynamically randomize the friction coefficient $\mu$ during rollouts to simulate abrupt transitions between terrain surfaces.

Figure 4 and Figure 10 visualize the ground reaction forces (GRFs) alongside the corresponding friction cone boundaries as the robot traverses terrains with varying friction. The induction of the proposed constraint embedding mechanism effectively achieves strong constraint satisfaction and zero-shot adaptation to dynamic safety requirements. More details are provided in Appendix H.

### 5.5. Ablation Study

We conduct extensive ablation studies to evaluate the critical components of PolyFlow (detailed results are provided in Appendix I).

**Necessity of constraint embedding.** To clarify the role of the constraint encoding block, we compared our default cross-attention architecture with an MLP-based encoder and a baseline without explicit constraint embedding (w/o). For tasks with static constraints (e.g., Maze2d, HalfCheetah), the "w/o" baseline performs surprisingly well, achieving performance comparable to other architectures. We hypothesize that this is because the model implicitly learns the

fixed safety boundaries from the state distribution. However, in the Unitree Go2 task, where constraints are highly state-dependent and time-varying, explicit constraint encoding becomes essential, and the "w/o" baseline suffers from a significant performance drop.

**Coupling between weight predictor and direction predictor.** We investigate whether the weight predictor $\gamma_\theta$ needs to be explicitly conditioned on the output of the direction predictor $d_\theta$. Comparing our decoupled architecture against a coupled variant shows only marginal differences in overall performance. This suggests that the weight predictor can implicitly infer directional context from the shared latent state and constraint embeddings without explicit direction input.

**Ray shooting operator.** We compare the standard exact minimization (Hard) operation in ray shooting against smoothed approximations (Softmin and Boltzmann). The results demonstrate that the direct exact minimization yields superior performance and strict safety, as it provides precise boundary estimations which are crucial when optimal actions lie on constraint boundaries.

**OT-based batch coupling.** Incorporating Optimal Transport (OT) data coupling within training batches further enhances rollout performance. This empirical finding aligns with Theorem 4.4, suggesting that OT matching effectively reduces the variance of the approximation error induced by discretization.

**Integration steps.** We evaluate PolyFlow across varying sample steps. PolyFlow maintains high reward returns even in extremely low sampling steps ($N = 2$). Crucially, PolyFlow maintains perfect safety across all step sizes, indicating strong safety robustness.

## 6. Conclusion

In this work, we introduce PolyFlow, a sample-efficient flow matching framework designed to provide strict safety guarantees for constrained generative modeling. By reformulating flow dynamics in discrete time and incorporating a projection-free architecture, the proposed method eliminates numerical discretization errors and the computational burden of post-hoc projections.

While PolyFlow demonstrates strong performance, it is subject to several limitations. Currently, the framework requires constraints to be formulated as convex polytopes. Additionally, it necessitates that the initial distribution resides within the feasible region, which incurs additional computational overhead during sampling. Further generalizing PolyFlow to non-convex constraint geometries—such as through convex decomposition strategies—remains an important and promising direction for future work.

## Acknowledgements

This work is supported by New Generation Artificial Intelligence-National Science and Technology Major Project (No. 2025ZD0122901) and the National Natural Science Foundation of China (Grant No. 62373242).

## Impact Statement

This paper presents work whose goal is to advance the field of Machine Learning. There are many potential societal consequences of our work, none which we feel must be specifically highlighted here.

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

## A. Preliminary: Projection-Free Constrained Optimization

Given a compact convex safety set $\mathcal{X}_c \subset \mathbb{R}^d$, our objective is to ensure that the final generated sample $x_1$ resides within this safe region. This can be formulated as a constrained optimization problem:

$$x^* = \min_{x \in \mathcal{X}_c} f(x). \tag{14}$$

Classic projection-based methods, which alternate between gradient descent and Euclidean projection, become computationally infeasible in high-dimensional spaces due to the prohibitive cost of the projection operator. This bottleneck motivates the use of projection-free approaches, most notably the Frank-Wolfe (FW) algorithm (Frank & Wolfe, 1956).

This algorithm circumvents the projection step by accessing the constraint set $\mathcal{X}_c$ solely through a Linear Minimization Oracle (LMO). The LMO solves a linear subproblem to identify an extreme point (vertex) $v^*$ that minimizes the inner product with the current gradient $g$:

$$v^* = \arg\min_{v \in \mathcal{X}_c} \langle g, v \rangle. \tag{15}$$

Geometrically, $v^*$ represents the vertex with the maximum projection along the negative gradient direction. The algorithm then updates the iterate along the direction $d^k = v^* - x^k$:

$$x^{k+1} = x^k + \gamma^k d^k = (1 - \gamma^k)x^k + \gamma^k v^*, \tag{16}$$

where $\gamma^k \in (0, 1)$ is the step size. By convexity of $\mathcal{X}_c$, the convex combination guarantees $x^{k+1} \in \mathcal{X}_c$. Consequently, the solution $x^k$ can be expressed as a sparse convex combination of the initial point and the history of the discovered vertices: $x^k = w^0 x^0 + \sum_{i=1}^{k} w^i v^i$.

However, the standard FW algorithm suffers from the monotonic expansion of the *active set* (the set of vertex representing the solution), leading to memory inefficiency and the "zig-zagging" phenomenon that hinders convergence. To mitigate this, Away-step FW(Lacoste-Julien & Jaggi, 2015) incorporates an active set maintenance mechanism. This involves dynamically adding new promising atoms while pruning suboptimal ones (via away-steps, for instance), thereby keeping the representation sparse. The state is thus parameterized by a dynamic vertex set $\mathcal{V}_k$:

$$\mathcal{V}_k = \text{Update}(\mathcal{V}_{k-1}), \tag{17}$$

$$x^k = x^0 + \sum_{v \in \mathcal{V}_k} w_v v, \quad \text{s.t.} \sum_{v \in \mathcal{V}_k} w_v = 1, \ w_v > 0. \tag{18}$$

## B. Theorem Proofs

In this section, we provide the detailed derivation of the discretization error bound presented in Theorem 4.4. We aim to quantify the deviation between the approximate distribution $\tilde{p}_t$ generated by the marginal discrete vector field and the true marginal distribution $p_t$. We establish a recursive upper bound on the error using the 2-Wasserstein distance.

### B.1. Proof of Theorem 4.4

First, we formally define the 2-Wasserstein distance.

**Definition B.1** (Wasserstein-2 Distance). Let $\mathcal{P}_2(\mathbb{R}^d)$ be the space of probability measures on $\mathbb{R}^d$ with finite second moments. For any two distributions $\mu, \nu \in \mathcal{P}_2(\mathbb{R}^d)$, the Wasserstein-2 distance is defined as:

$$W_2(\mu, \nu) \triangleq \left( \inf_{\pi \in \Pi(\mu, \nu)} \int_{\mathbb{R}^d \times \mathbb{R}^d} \|x - y\|^2 d\pi(x, y) \right)^{1/2}, \tag{19}$$

where $\Pi(\mu, \nu)$ denotes the set of all joint distributions (couplings) with marginals $\mu$ and $\nu$. By definition, for any specific coupling $(X, Y) \sim \pi$, inequality $W_2(\mu, \nu) \leq \sqrt{\mathbb{E}[\|X - Y\|^2]}$ holds.

**Dynamics Setup.** Consider the transition from time step $t$ to $t + 1$.

- **Reference Process:** Follows the conditional vector field $u_t(x|z)$.

$$x_{t+1} = x_t + u_t(x_t|z), \quad x_t \sim p_t, \ z \sim p(z|x_t). \tag{20}$$

- **Approximate Process:** Follows the marginal expectation field $\tilde{u}_t(x) = \mathbb{E}_{z|x}[u_t(x|z)]$.

$$\tilde{x}_{t+1} = \tilde{x}_t + \tilde{u}_t(\tilde{x}_t), \quad \tilde{x}_t \sim \tilde{p}_t. \tag{21}$$

To derive the recursive bound, we first decompose the error into "transmission error" and "matching error" using Minkowski's inequality.

**Lemma B.2** (Error Decomposition). *Let $L^2(\Omega)$ denote the space of random variables with finite second moments. For random variables $x_{t+1}$ and $\tilde{x}_{t+1}$ at time $t+1$, their $L^2$ distance satisfies:*

$$\sqrt{\mathbb{E}\|x_{t+1} - \tilde{x}_{t+1}\|^2} \leq \sqrt{\mathbb{E}\|\mathcal{T}(x_t) - \mathcal{T}(\tilde{x}_t)\|^2} + \sqrt{\mathbb{E}\|u_t(x_t|z) - \tilde{u}_t(x_t)\|^2}, \tag{22}$$

*where we define the transition mapping $\mathcal{T}(x) \triangleq x + \tilde{u}_t(x)$.*

*Proof.* We construct the difference between $x_{t+1}$ and $\tilde{x}_{t+1}$:

$$x_{t+1} - \tilde{x}_{t+1} = (x_t + u_t(x_t|z)) - (\tilde{x}_t + \tilde{u}_t(\tilde{x}_t)) \tag{23}$$

$$= \underbrace{(x_t + \tilde{u}_t(x_t)) - (\tilde{x}_t + \tilde{u}_t(\tilde{x}_t))}_{A} + \underbrace{(u_t(x_t|z) - \tilde{u}_t(x_t))}_{B}. \tag{24}$$

Here, term $A$ represents the discrepancy caused by the deterministic map $\mathcal{T}$, and term $B$ represents the difference between the conditional field and the marginal field. Applying Minkowski's inequality (the triangle inequality for the $L^2$ norm, $\|A + B\|_2 \leq \|A\|_2 + \|B\|_2$):

$$\sqrt{\mathbb{E}\|x_{t+1} - \tilde{x}_{t+1}\|^2} \leq \sqrt{\mathbb{E}\|A\|^2} + \sqrt{\mathbb{E}\|B\|^2}. \tag{25}$$

Substituting the definitions of $A$ and $B$ concludes the proof. □

**Lemma B.3** (Lipschitz Transmission Bound). *Assume the marginal expectation field $\tilde{u}_t(x)$ is $L$-Lipschitz continuous, i.e., $\|\tilde{u}_t(x) - \tilde{u}_t(y)\| \leq L\|x - y\|$. Then the map $\mathcal{T}(x) = x + \tilde{u}_t(x)$ is $(1 + L)$-Lipschitz, and:*

$$\sqrt{\mathbb{E}\|\mathcal{T}(x_t) - \mathcal{T}(\tilde{x}_t)\|^2} \leq (1 + L)\sqrt{\mathbb{E}\|x_t - \tilde{x}_t\|^2}. \tag{26}$$

*Proof.* Using the triangle inequality and the Lipschitz assumption:

$$\|\mathcal{T}(x_t) - \mathcal{T}(\tilde{x}_t)\| = \|(x_t - \tilde{x}_t) + (\tilde{u}_t(x_t) - \tilde{u}_t(\tilde{x}_t))\| \tag{27}$$

$$\leq \|x_t - \tilde{x}_t\| + \|\tilde{u}_t(x_t) - \tilde{u}_t(\tilde{x}_t)\| \tag{28}$$

$$\leq \|x_t - \tilde{x}_t\| + L\|x_t - \tilde{x}_t\| \tag{29}$$

$$= (1 + L)\|x_t - \tilde{x}_t\|. \tag{30}$$

Squaring both sides, taking the expectation, and applying the square root yields the result. □

Based on the lemmas above, we now prove the recursive upper bound for the Wasserstein error.

**Theorem 4.4** (Restated). *Assume that the marginal expectation field $\tilde{u}_t(x)$ is $L$-Lipschitz continuous. Let $W_2(p_t, \tilde{p}_t) = \sqrt{\mathbb{E}\|x_t - \tilde{x}_t\|^2}$ be the 2-Wasserstein distance between the true marginal $p_t$ and the approximate marginal $\tilde{p}_t$. Then at step $t + 1$:*

$$W_2(p_{t+1}, \tilde{p}_{t+1}) \leq (1 + L)W_2(p_t, \tilde{p}_t) + \sqrt{\mathbb{E}_z \mathbb{E}_{x_t \sim p_{t|Z}}[\|\tilde{u}_t(x_t) - u_t(x_t|z)\|^2]}. \tag{31}$$

*Proof.* 1. **Construct Optimal Coupling:** Assume that at time $t$, the pair of random variables $(x_t, \tilde{x}_t)$ follows the optimal coupling between $p_t$ and $\tilde{p}_t$, such that $\sqrt{\mathbb{E}\|x_t - \tilde{x}_t\|^2} = W_2(p_t, \tilde{p}_t)$.

2. **Apply Error Decomposition:** Let $(x_{t+1}, \tilde{x}_{t+1})$ evolve according to the dynamics described in the definitions. By the definition of the Wasserstein distance (as the infimum over all couplings):

$$W_2(p_{t+1}, \tilde{p}_{t+1}) \leq \sqrt{\mathbb{E}\|x_{t+1} - \tilde{x}_{t+1}\|^2}. \tag{32}$$

Using Lemma B.2, we expand the right-hand side:

$$W_2(p_{t+1}, \tilde{p}_{t+1}) \leq \sqrt{\mathbb{E}\|\mathcal{T}(x_t) - \mathcal{T}(\tilde{x}_t)\|^2} + \sqrt{\mathbb{E}\|u_t(x_t|z) - \tilde{u}_t(x_t)\|^2}. \tag{33}$$

3. **Apply Lipschitz Bound:** Substituting Lemma B.3 into the first term:

$$W_2(p_{t+1}, \tilde{p}_{t+1}) \leq (1+L)\sqrt{\mathbb{E}\|x_t - \tilde{x}_t\|^2} + \sqrt{\mathbb{E}_z \mathbb{E}_{x_t \sim p_{t|Z}}[\|\tilde{u}_t(x_t) - u_t(x_t|z)\|^2]} \tag{34}$$

$$= (1+L)W_2(p_t, \tilde{p}_t) + \sqrt{\mathbb{E}_z \mathbb{E}_{x_t \sim p_{t|Z}}[\|\tilde{u}_t(x_t) - u_t(x_t|z)\|^2]}. \tag{35}$$

The final equality holds by the assumption of the optimal coupling at time $t$. Note that the second term represents the inherent variance of the conditional vector field, which corresponds to the irreducible error of the flow matching objective. $\square$

### B.2. Proof of Theorem 4.5

**Theorem 4.5 (Restated).** Let $\mathcal{X}_c$ be a convex set. If the conditional flow is interior safe (i.e., $x + u_t(x|z) \in \mathcal{X}_c$ for all $x \in \mathcal{X}_c, z$), then the marginal expectation field $u_t^p(x)$ is also interior safe.

*Proof.* By the definition of marginal expectation field $u_t^p(x)$, the next state in the marginal discrete flow, denoted as $\tilde{x}_{t+1}$, is given by:

$$\tilde{x}_{t+1} = x_t + u_t^p(x_t) = x_t + \mathbb{E}_{z \sim p_{t|Z}(\cdot|z)}[u_t(x_t|z)]. \tag{36}$$

Using the linearity of the expectation operator, we can move the deterministic term $x_t$ inside the expectation:

$$\tilde{x}_{t+1} = \mathbb{E}_{z \sim p_{Z|t}(\cdot|x_t)}[x_t + u_t(x_t|z)]. \tag{37}$$

Let $\omega_t(z) = x_t + u_t(x_t|z)$. According to the assumption of conditional interior safety, we have $\omega_t(z) \in \mathcal{X}_c$ for all $z$ in the support of the posterior distribution $p_{Z|t}(\cdot|x_t)$. Since $\mathcal{X}_c$ is a convex set, it is closed under convex combinations (and by extension, expectations). Therefore, the expectation of the random variable $\omega_t(z)$, which takes values strictly within $\mathcal{X}_c$, must also reside within $\mathcal{X}_c$:

$$\tilde{x}_{t+1} = \mathbb{E}_{z \sim p_{Z|t}}[\omega_t(z)] \in \mathcal{X}_c.$$

Hence, for any $x_t \in \mathcal{X}_c$, the marginal update $\tilde{x}_{t+1}$ remains in $\mathcal{X}_c$, which proves that the marginal flow is interior safe. $\square$

## C. Ray Shooting Operator Details

The Ray Shooting operator $\text{RS}(x, d) : \mathcal{X}_c \times \mathbb{R}^d \to \partial \mathcal{X}_c$ computes the intersection point between a ray originating at $x$ with direction $d$ and the boundary of the convex polytope $\mathcal{X}_c$.

Let the safety polytope be defined by $m$ linear inequality constraints:

$$\mathcal{X}_c = \{z \in \mathbb{R}^n \mid a_i^\top z \leq b_i, \; \forall i \in \{1, \ldots, m\}\}, \tag{38}$$

where $a_i \in \mathbb{R}^n$ and $b_i \in \mathbb{R}$ define the normal vector and offset of the $i$-th hyperplane, respectively.

Given a current feasible state $x_t \in \mathcal{X}_c$ and a predicted direction vector $d \in \mathbb{R}^n$, the ray can be parameterized as $r(\lambda) = x_t + \lambda d$ for $\lambda \geq 0$. To find the intersection with the boundary $\partial \mathcal{X}_c$, we seek the maximum step size $\lambda^*$ such that the point remains feasible.

Substituting the ray equation into the constraints:

$$a_i^\top(x_t + \lambda d) \leq b_i \implies \lambda(a_i^\top d) \leq b_i - a_i^\top x_t. \tag{39}$$

Let $s_i = b_i - a_i^\top x_t$ denote the "slack" of the $i$-th constraint at the current state. Since $x_t$ is feasible, $s_i \geq 0$. We analyze the term $v_i = a_i^\top d$:

- If $v_i \leq 0$, the ray is moving parallel to or away from the boundary of the $i$-th constraint. Since the constraint is currently satisfied, moving in this direction will never violate it.

---

**Algorithm 2** PolyFlow Sampling (Inference)

---

1: **Input:** Initial sample $x_0 \sim p$, Trained models $\theta$.
2: **Constraint:** Safety Polytope $\mathcal{X}_c$.
3: **for** $t = 0$ **to** $T - 1$ **do**
4:     $d_t \leftarrow d_\theta(x_t, t, \mathcal{X}_c)$
5:     $\gamma_t \leftarrow \gamma_\theta(x_t, t, \mathcal{X}_c)$
6:     $z_t^* \leftarrow \text{RS}(x_t, d_t, \mathcal{X}_c)$                            ▷ *Find valid boundary target*
7:     $x_{t+1} \leftarrow x_t + \gamma_t \cdot (z_t^* - x_t)$
8:     *// Note:* $x_{t+1} \in \mathcal{X}_c$ *is guaranteed by convexity.*
9: **end for**
10: **Return** generated sample $x_T$

---

- If $v_i > 0$, the ray is moving towards the boundary. The constraint will be violated if $\lambda > s_i / v_i$.

Since the polytope is compact, the ray must eventually intersect at least one boundary. The intersection occurs at the smallest positive $\lambda$ imposed by any constraint moving towards the boundary:

$$\lambda^* = \min_{\{i \mid a_i^\top d > 0\}} \left( \frac{b_i - a_i^\top x_t}{a_i^\top d} \right). \tag{40}$$

Finally, the intersection point $z^*$ is computed as:

$$z^* = \text{RS}(x_t, d) = x_t + \lambda^* d. \tag{41}$$

**Differentiability:** The scalar $\lambda^*$ is the minimum of a finite set of rational functions. The operation is differentiable almost everywhere with respect to both $x_t$ and $d$, except at the specific manifold where the active constraint set changes (i.e., when the ray hits exactly the intersection of two faces). In deep learning practice, this behaves similarly to the `max` or ReLU operations, allowing gradients to propagate back through the parameters of the active constraint $k$ (where $k = \arg\min(\dots)$) to update the direction network $d_\theta$. The gradient is given by:

$$\nabla_d z^* = \lambda^* I + d(\nabla_d \lambda^*)^\top, \quad \text{where } \nabla_d \lambda^* = -\frac{\lambda^*}{a_k^\top d} a_k. \tag{42}$$

This ensures that the entire PolyFlow framework remains fully differentiable and end-to-end trainable.

## D. Baselines

- **Flow**: We adopt an unconstrained flow matching model trained along the optimal transport path. To ensure architectural consistency across methods, all flow-based models in our experiments (including **Flow**, **FlowTrunc**, **SafeFlow**, and **GaugeFlow**) share DiT (Peebles & Xie, 2023) as the network backbone.

- **FlowTrunc**: This baseline is a direct constrained variant of Flow, which enforces constraints through a hard projection strategy during sampling. Specifically, after each numerical integration step, any trajectory point that violates the safety constraints is projected back onto the feasible region.

- **SafeFlow** (Dai et al., 2025): This method represents a state-of-the-art approach among CBF-based flow models. During the sampling process, it introduces an auxiliary control term at each step to enforce safety constraints, which requires solving QP problems When more than two CBF constraints are present. In our implementation, the `qpth` library is used to accelerate QP solving via parallelization.

- **SafeDiffuser** (Xiao et al., 2023): While its constraint enforcement mechanism is conceptually similar to that of SafeFlow, Safediffuser is fundamentally built upon diffusion models rather than flow matching. It includes three variants: RoSD, ReSD, and TVSD. In this work, We directly adopt the official model architectures and hyperparameters released by the authors.

- **GaugeFlow** (Li et al., 2025b): This method employs a gauge transformation that maps the convex feasible region to a unit ball, within which flow matching is performed. Points that deviate from the unit ball are corrected via a reflection mechanism. GaugeFlow requires a fixed interior point within the feasible region; here, we select the center of the Chebyshev ball. Due to its inherent limitation to static constraints, GaugeFlow is evaluated exclusively on the Gym tasks.

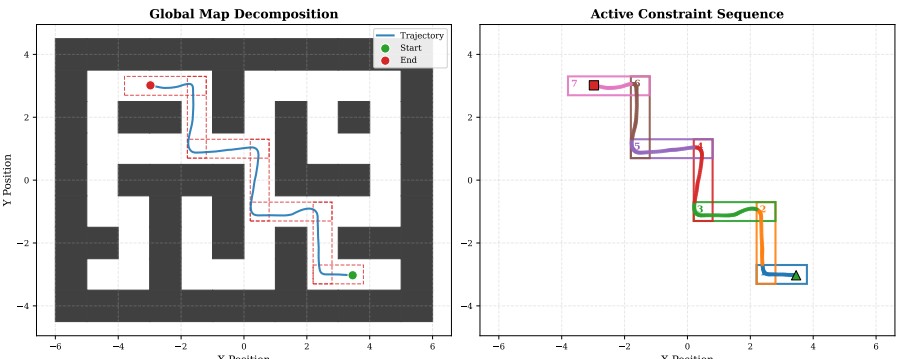

*Figure 6.* **Visualization of geometric constraint decomposition**. ( Left: global maze environment showing the reference trajectory (blue) from start (green) to goal (red). The free space is decomposed into a set of maximal empty rectangles (dashed outlines). Right: the resulting sequence of active linear constraints derived via the greedy allocation strategy. The trajectory is segmented into distinct phases (labeled 1-7), with each segment confined to a specific maximal rectangle (solid colored boxes) that maximizes the covered future horizon.

## E. Experiment Metrics.

We evaluate the proposed method and all baselines from three primary perspectives: safety satisfaction, distribution matching fidelity, and trajectory smoothness. The evaluation metrics are defined as follows:

- **Safety Rate (R):** Safety Rate measures the proportion of generated trajectories that strictly satisfy all safety constraints. Formally, $R$ is defined as the ratio between the number of feasible trajectories and the total number of generated trajectories. A value of $R = 1$ indicates perfect constraint satisfaction.

- **Maximum Mean Discrepancy (MMD):** To evaluate the distribution mismatch, we compute the MMD using a Gaussian radial basis function (RBF) kernel. The kernel bandwidth $\sigma$ is determined dynamically using the median heuristic on the combined distance matrix of generated and ground-truth samples.

- **Wasserstein Distance ($W_2$):** We measure the geometric discrepancy between generated and ground-truth distributions using the 2-Wasserstein distance. The distance is computed via the POT library (Flamary et al., 2021), with the squared Euclidean distance as the cost metric.

- **KL Divergence (KL):** We approximate the Kullback-Leibler divergence $D_{KL}(P_{\text{true}}||Q_{\text{gen}})$ using Gaussian Kernel Density Estimation (KDE). To ensure numerical stability and prevent singular matrices during density evaluation, a small Gaussian jitter ($\epsilon = 10^{-5}$) is added to the data.

- **Curvature Smoothness (Cur):** This metric quantifies the average rate of change in curvature along the generated trajectories. Lower values indicate smoother geometrically paths.

- **Acceleration Smoothness (Acc):** Acceleration Smoothness measures the average rate of change in acceleration (analogous to jerk in dynamical systems). Lower values correspond to smoother trajectory dynamics.

- **TotalTime:** This metric records the wall-clock time required to generate a full batch of trajectories. To ensure fair comparison and reduce measurement variance, all methods are evaluated with a fixed batch size of 200. Consequently, TotalTime reports the average duration required to generate a batch of 200 samples.

- **StepTime:** This represents the average computation time per sampling step. Since different methods may utilize varying numbers of integration steps, **TotalTime** serves as the primary indicator for comparing the overall inference efficiency.

# F. Safe Planning in Maze Environment

**Specifications**    The maze layout used in this experiment is illustrated in Figure 3.   Its geometry follows the `PointMaze_Large-v3` environment from the gymnasium-robotics library (de Lazcano et al., 2024), with one critical modification: **the width of all corridors is narrowed by** $0.4\,\text{m}$. This adjustment increases the difficulty of the navigation task by tightening the geometric constraints and increasing the risk of collision. To facilitate controlled comparisons across methods, the start position is fixed at the bottom-right corner of the maze, while the goal position is fixed at the top-left corner. To introduce variability, we apply a uniform perturbation of range $[-0.25, 0.25]$ m to both the start and goal coordinates. The objective is to generate collision-free paths connecting the start to the goal positions.

Different methods require different formulations of safety constraints. For CBF-based baselines (e.g., SafeFlow, RoSD), manually designing valid CBFs for complex environments remains a significant challenge. This difficulty arises from two factors: (1) CBF methods theoretically require a continuous superlevel set to define the feasible region, whereas physical boundaries in maze environments are typically discontinuous, necessitating approximation; (2) The safety control set is theoretically guaranteed to be non-empty only under single constraints; as the number of constraints increases (e.g., multiple walls), the intersection of safe sets may become empty, requiring the introduction of slack variables. In our experiment, we approximate the rectangular obstacles using 4th-order super-ellipses to ensure differentiability:

$$\left(\frac{x - x_c}{a}\right)^4 + \left(\frac{y - y_c}{b}\right)^4 - r \geq 0, \tag{43}$$

where $(x_c, y_c)$ denotes the center of the obstacle, $a$ and $b$ represent the semi-axis lengths, and $r$ is a scaling radius. We defined $N$ such CBFs to represent the maze boundaries, as illustrated in Figure 3.

For our PolyFlow method, strict safety relies on convex polytope feasible regions. Given that the global free space of the maze is non-convex, we employ a local convex decomposition strategy. We decompose the free space into a set of Maximal Empty Rectangles (MERs). Subsequently, we assign each trajectory point in the dataset to a specific local constraint by greedily identifying the nearest feasible rectangle (See Algorithm 3). Consequently, every point in the dataset is associated with a local rectangular constraint, as shown in Figure 6.

**Dataset Generation.**    We constructed a demonstration policy using a Q-Iteration algorithm coupled with a PD controller in the *original* `PointMaze_Large-v3` environment. Trajectory data were collected by rolling out this policy. To enhance dataset diversity, minor random perturbations were injected into the policy action during data collection. In total, we collected 1,000 trajectories for training. It is crucial to note that the demonstration policy was trained on the original map configuration, without the corridor narrowing applied in our experimental setting. As a result, the training dataset inherently contains trajectories that violate the stricter safety constraints of the test environment, serving as a robust test for learning safe generation from potentially unsafe demonstrations.

**Training Details.**    All models were trained on a single NVIDIA GeForce RTX 4090 GPU. The specific hyperparameters for our method and all baseline algorithms are summarized in Table 4. Specifically, the model architecture and training settings for the RoSD method were adopted directly from the configurations in (Xiao et al., 2023).

# G. Safe Locomotion in Gym Environment

In control tasks, we formulate the generative model as a receding-horizon controller. At each control step $t$, the model is conditioned on the current observation $o_t$ and predicts the joint state-action trajectory over a future horizon of length $T$: $\tau_{pred} = (a_t, o_{t+1}, a_{t+1}, \ldots, o_{t+T-1}, a_{t+T-1})$. A key requirement of our framework is that all predicted states along this generated trajectory strictly satisfy the predefined safety constraints. We utilize three representative environments from the Gymnasium library (Towers et al., 2024) as our experimental testbed, including `Hopper-v5`, `Walker2d-v5`, and `HalfCheetah-v5`.

## G.1. Specifications

**Hopper Task**    To comprehensively evaluate the safety satisfaction of the proposed method, two constrained variants of the Hopper environment are considered:

---

**Algorithm 3** Trajectory-Constraint Decomposition via Maximal Empty Rectangles

---

1: **Input:** Maze Map $\mathcal{M}$, Trajectory $\tau = \{s_0, s_1, \ldots, s_T\}$ where $s_t \in \mathbb{R}^2$
2: **Output:** Sequence of Linear Constraints $\{(A_t, b_t)\}_{t=0}^T$
3: $\mathcal{R}_{\max} \leftarrow$ FINDMAXIMALRECTANGLES$(\mathcal{M})$ ▷*Pre-process: Find all maximal empty rectangles in the grid map*
4: Initialize time index $t \leftarrow 0$
5: **while** $t < T$ **do**
6:     $\mathcal{C}_t \leftarrow \{R \in \mathcal{R}_{\max} \mid s_t \in R\}$ ▷*Find all valid rectangles containing current state $s_t$*
7:     **if** $\mathcal{C}_t = \emptyset$ **then**
8:         $R^* \leftarrow$ FindNearestRect$(s_t, \mathcal{R}_{\max})$ ▷*Handle outliers*
9:         $k^* \leftarrow t + 1$
10:     **else**
11:         # *Greedy Selection: Choose rect that covers the longest future horizon*
12:         $k_{\text{best}} \leftarrow -1$
13:         $R^* \leftarrow$ null
14:         **for all** $R \in \mathcal{C}_t$ **do**
15:             Find largest $k \in [t, T]$ such that $\forall j \in [t, k], s_j \in R$
16:             **if** $k > k_{\text{best}}$ **then**
17:                 $k_{\text{best}} \leftarrow k$
18:                 $R^* \leftarrow R$
19:             **end if**
20:         **end for**
21:         $k^* \leftarrow k_{\text{best}}$
22:     **end if**
23:     $A^*, b^* \leftarrow$ CalculateConstraints$(R^*)$
24:     # *Assign constraints to the sub-sequence*
25:     **for** $j = t$ **to** $k^* - 1$ **do**
26:         $A_j \leftarrow A^*$
27:         $b_j \leftarrow b^*$
28:     **end for**
29:     $t \leftarrow k^*$ ▷ *Jump to the end of the covered segment*
30: **end while**
31: **return** $\{(A_t, b_t)\}_{t=0}^T$

---

- **Hopper-Simple**: Following (Xiao et al., 2023), we impose a dynamic torso height constraint:

$$z + 0.01 \cdot v_z \leq 1.6, \tag{44}$$

  where $z$ and $v_z$ denote the vertical position and velocity of the hopper, respectively.

- **Hopper-Hard**: This variant introduces a stricter set of composite constraints to narrow the feasible region, including the dynamic height limit, a lower bound on height, and velocity bounds:

$$z + 0.01 \cdot v_z \leq 1.5, \quad z \geq 0.8, \quad -2.5 \leq v_z \leq 2.5. \tag{45}$$

Notably, due to the coupling between position and velocity in the dynamic constraint ($z + 0.01 \cdot v_z$), valid states cannot be restored via simple truncation or clamping operations. This characteristic makes these tasks challenging benchmarks for evaluating safe generation capabilities.

**Walker2d Task**     Similar to the Hopper tasks, we define two Walker2d variants with increasing constraint complexity to evaluate the adaptability of PolyFlow:

- **Walker2d-Simple**: We impose a dynamic upper bound on the torso height:

$$z + 0.01 \cdot v_z \leq 1.35, \tag{46}$$

  where $z$ and $v_z$ denote the vertical position and velocity of the torso, respectively.

- **Walker2d-Hard**: This variant introduces a stricter set of composite constraints, including the dynamic height limit, a minimum height requirement, and bounded vertical velocity:

$$z + 0.01 \cdot v_z \leq 1.35, \quad z \geq 0.9, \quad -1.4 \leq v_z \leq 1.4. \tag{47}$$

**HalfCheetah Task**  To evaluate safety under high-dimensional action spaces, we untilize the `HalfCheetah-v5` environment and impose additional linear constraints on the action space, specifically targeting actuator saturation and mechanical torsion. The constraints are defined as:

$$\textbf{Actuator Saturation:} \quad u_0 + u_1 \leq 1.2, \quad u_3 + u_4 \leq 1.2 \tag{48}$$
$$\textbf{Anti-Torsion:} \quad |u_0 - u_3| \leq 0.8 \tag{49}$$

Here, $u_0$ and $u_1$ denote the torque outputs applied to the *thigh* and *knee* joints of the back leg, respectively, while $u_3$ and $u_4$ correspond to the torque outputs for the *thigh* and *knee* joints of the front leg.

These constraints reflect fundamental physical limitations of robotic systems:

- **Actuator Saturation Constraint** (Eq. (48)) limits the combined torque exerted by the thigh and knee actuators of each leg. Applying large torques simultaneously in the same direction can induce excessive mechanical stress and actuator overheating.

- **Anti-Torsion Constraint** (Eq. (49)) restricts the differential torque between the primary drive joints (the back and front thighs). Excessive disparity between these torques can generate harmful torsional forces on the torso, potentially compromising structural integrity and locomotion stability in high-speed scenarios.

### G.2. Datasets

We use training datasets from the Minari library (Younis et al., 2024), specifically `mujoco/hopper/medium-v0`, `mujoco/walker2d/medium-v0`, and `mujoco/halfcheetah/medium-v0`. Each dataset consists of trajectories collected by policies with medium-level performance that were unaware of the safety constraints defined for our tasks. As a result, a portion of the trajectories in the datasets explicitly violate the safety boundaries, providing a realistic and challenging setting for learning safe control from imperfect demonstrations.

### G.3. Training Details

All models were trained on a single NVIDIA GeForce RTX 4090 GPU. The specific hyperparameters for our method and all baseline algorithms are summarized in Table 5, Table 6, and Table 7 for the Hopper, Walker2d, and HalfCheetah environments, respectively. Specifically, the model architecture and training settings for the RoSD method were adopted directly from the configurations in (Xiao et al., 2023).

### G.4. Additional Experimental Results

We provide additional qualitative and quantitative results to further illustrate the behavior of different methods under safety constraints. Figure 7 visualizes the closed-loop rollout trajectories in the Hopper environment. Although PolyFlow, SafeFlow, and RoSD enforce strict constraint satisfaction during open-loop trajectory generation, constraint violations may still arise during closed-loop execution due to the mismatch between predicted trajectories and environment dynamics. Figures 8 and 9 visualize the distributions of generated states or actions in the Walker2d and HalfCheetah tasks. Compared to unconstrained baselines(Flow), methods that explicitly incorporate safety constraints produce samples that are more tightly concentrated within feasible regions. Consistent with these observations, the detailed generation and rollout metrics reported in Tables 8 and 9 also indicate that constraint-aware methods consistently improve safety compared to the unconstrained flow baseline. Among these approaches, PolyFlow achieves the most favorable trade-off, delivering the strongest safety performance while preserving competitive distributional quality and computational efficiency across all environments.

*Table 4.* Hyperparameter settings for different methods in the Maze2d environment.

(a) Flow / SafeFlow

| Parameter | Value |
|---|---|
| Hidden dim. | 128 |
| Head num. | 4 |
| DiT block num. | 4 |
| Batch size | 100 |
| Optimizer | Adam |
| LR | 1e-4 |
| Train steps | 1e4 |
| Sample Steps | 200 |
| Sequence Length | 300 |

(b) RoSD

| Parameter | Value |
|---|---|
| Unet layers | 3 |
| Hidden dim. | [32, 128, 256] |
| Batch size | 100 |
| Optimizer | Adam |
| LR | 1e-4 |
| Train steps | 1e4 |
| Diff. Steps | 256 |
| Sequence Length | 300 |

(c) PolyFlow (Ours)

| Parameter | Value |
|---|---|
| Hidden dim. | 128 |
| Head num. | 4 |
| Traj. enc. layers | 2 |
| Cons. enc. layers | 2 |
| Weight enc. layers | 2 |
| Batch size | 100 |
| Optimizer | Adam |
| LR | 1e-4 |
| Train steps | 1e4 |
| Discrete Steps | 10 |
| Sequence Length | 300 |

*Table 5.* Hyperparameter settings for different methods in the Hopper environment.

(a) Flow / SafeFlow

| Parameter | Value |
|---|---|
| Hidden dim. | 128 |
| Head num. | 4 |
| DiT block num. | 4 |
| Batch size | 64 |
| Optimizer | Adam |
| LR | 1e-4 |
| Train steps | 1e6 |
| Sample Steps | 200 |
| Horizon $T$ | 100 |

(b) RoSD

| Parameter | Value |
|---|---|
| Unet layers | 4 |
| Hidden dim. | [32, 64, 128, 256] |
| Batch size | 32 |
| Optimizer | Adam |
| LR | 2e-4 |
| Train steps | 1e6 |
| Diff. Steps | 20 |
| Horizon $T$ | 600 |

(c) PolyFlow (Ours)

| Parameter | Value |
|---|---|
| Hidden dim. | 128 |
| Head num. | 4 |
| Traj. enc. layers | 4 |
| Cons. enc. layers | 2 |
| Weight enc. layers | 2 |
| Batch size | 64 |
| Optimizer | Adam |
| LR | 1e-4 |
| Train steps | 1e6 |
| Discrete Steps | 10 |
| Horizon $T$ | 100 |

*Table 6.* Hyperparameter settings for different methods in the Walker2d environment.

(a) Flow / SafeFlow

| Parameter | Value |
|---|---|
| Hidden dim. | 128 |
| Head num. | 4 |
| DiT block num. | 4 |
| Batch size | 64 |
| Optimizer | Adam |
| LR | 1e-4 |
| Train steps | 1e6 |
| Sample Steps | 100 |
| Horizon $T$ | 100 |

(b) RoSD

| Parameter | Value |
|---|---|
| Unet layers | 4 |
| Hidden dim. | [64, 128, 256, 512] |
| Batch size | 32 |
| Optimizer | Adam |
| LR | 2e-4 |
| Train steps | 1e6 |
| Diff. Steps | 20 |
| Horizon $T$ | 160 |

(c) PolyFlow (Ours)

| Parameter | Value |
|---|---|
| Hidden dim. | 128 |
| Head num. | 4 |
| Traj. enc. layers | 4 |
| Cons. enc. layers | 2 |
| Weight enc. layers | 2 |
| Batch size | 64 |
| Optimizer | Adam |
| LR | 1e-4 |
| Train steps | 1e6 |
| Discrete Steps | 10 |
| Horizon $T$ | 100 |

*Table 7.* Hyperparameter settings for different methods in the HalfCheetach environment.

(a) Flow / SafeFlow

| Parameter | Value |
|---|---|
| Hidden dim. | 128 |
| Head num. | 4 |
| DiT block num. | 4 |
| Batch size | 64 |
| Optimizer | Adam |
| LR | 1e-4 |
| Train steps | 1e6 |
| Sample Steps | 100 |
| Horizon $T$ | 100 |

(b) RoSD

| Parameter | Value |
|---|---|
| Unet layers | 4 |
| Hidden dim. | [64, 128, 256, 512] |
| Batch size | 32 |
| Optimizer | Adam |
| LR | 2e-4 |
| Train steps | 1e6 |
| Diff. Steps | 20 |
| Horizon $T$ | 160 |

(c) PolyFlow (Ours)

| Parameter | Value |
|---|---|
| Hidden dim. | 128 |
| Head num. | 4 |
| Traj. enc. layers | 4 |
| Cons. enc. layers | 2 |
| Weight enc. layers | 2 |
| Batch size | 64 |
| Optimizer | Adam |
| LR | 1e-4 |
| Train steps | 1e6 |
| Discrete Steps | 10 |
| Horizon $T$ | 100 |

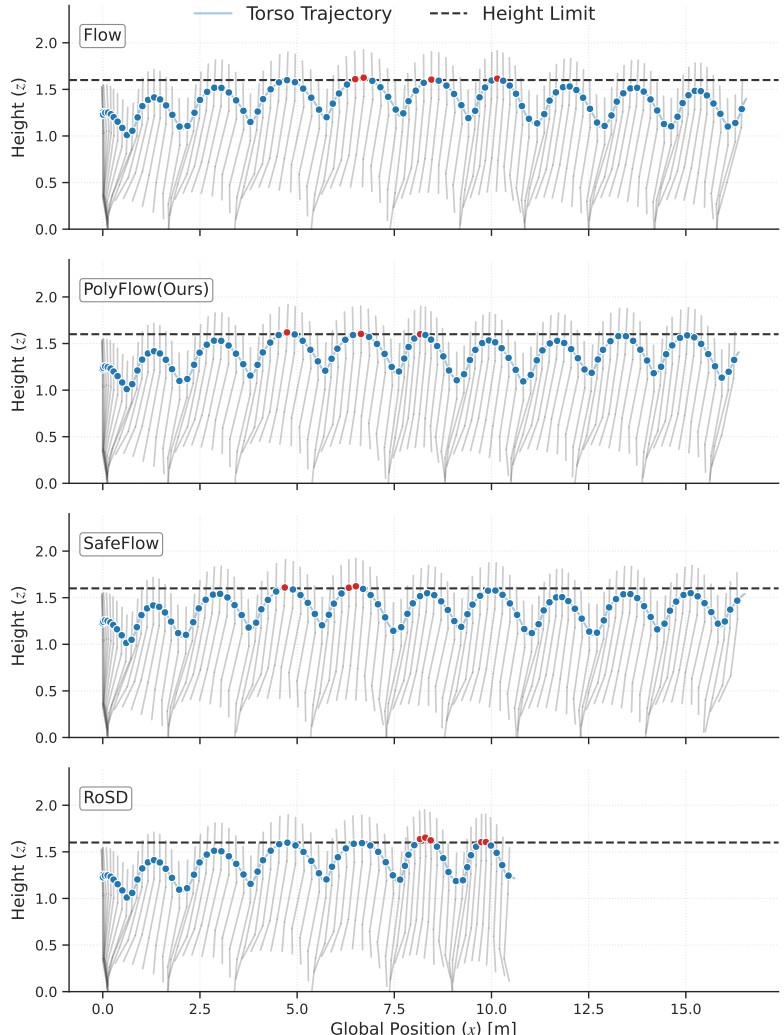

*Figure 7.* The sampled rollout trajectory of different methods in Hopper-Simple task.

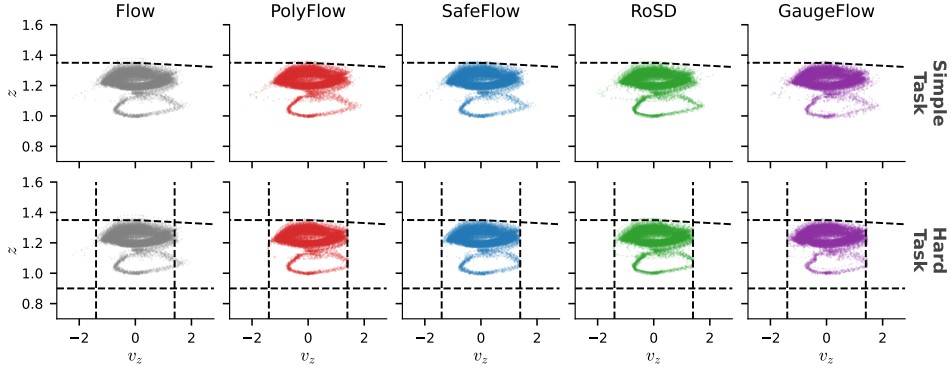

*Figure 8.* **Visualization of generated state samples on Walker2d tasks.** This figure illustrates the distribution of sampling points generated by different methods for the Walker2d-Simple and Walker2d-Complex tasks, plotted on the $(v_z, z)$ plane. The black dashed lines represent the constraint boundaries.

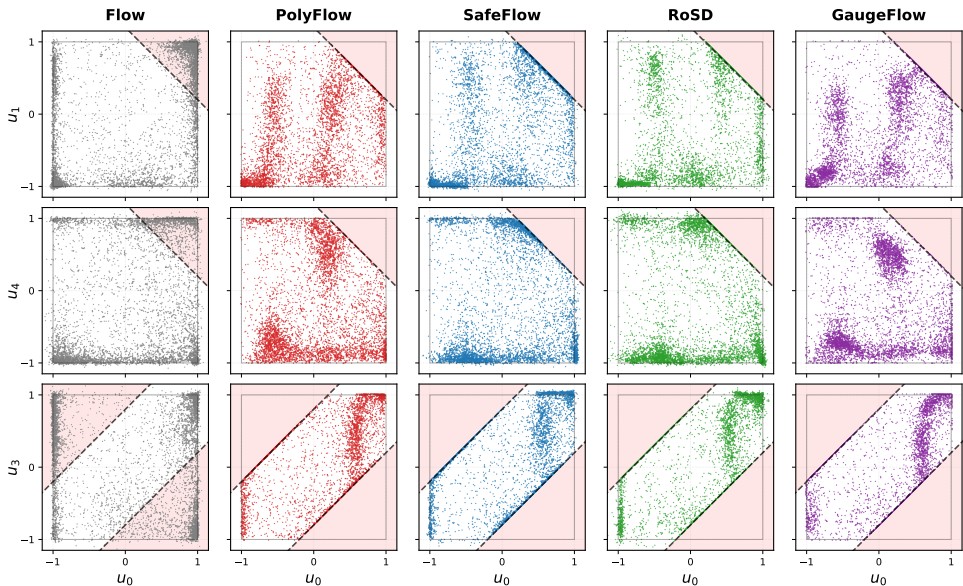

*Figure 9.* **Visualization of generated action samples on the HalfCheetah task.** The black dashed lines represent the constraint boundaries, while the red shaded regions indicate areas of constraint violation.

*Table 8.* Comparison of generation metrics across five different environments. **PolyFlow (Ours)** consistently achieves high safety and competitive performance metrics.

**(a)** Hopper-Simple

| Metric | Flow | PolyFlow | SafeFlow | RoSD | GaugeFlow |
|---|---|---|---|---|---|
| Safety(↑) | 0.755 | **1.000** | **1.000** | **1.000** | **1.000** |
| MMD(↓) | 2.10-4 | **1.74-4** | 2.10-4 | 1.97-3 | 2.41-4 |
| W(↓) | 1.152 | **1.114** | 1.152 | 1.788 | 1.159 |
| KL(↓) | 0.436 | **0.393** | 0.415 | 1.862 | 0.433 |
| Acc(↓) | **0.010** | 0.011 | **0.010** | 0.013 | 0.011 |
| TotalTime(↓) | 0.698 | **0.075** | 0.833 | 1.749 | 0.867 |
| StepTime(↓) | **0.003** | 0.007 | 0.004 | 0.087 | 0.004 |

**(b)** Hopper-Hard

| Metric | Flow | PolyFlow | SafeFlow | RoSD | GaugeFlow |
|---|---|---|---|---|---|
| Safety(↑) | 0.005 | **1.000** | **1.000** | **1.000** | **1.000** |
| MMD(↓) | 2.10-4 | **1.74-4** | 2.33-4 | 5.22-3 | 2.52-4 |
| W(↓) | 1.152 | **1.117** | 1.157 | 1.963 | 1.161 |
| KL(↓) | **0.436** | 0.553 | 0.608 | 23.894 | 0.592 |
| Acc(↓) | **0.010** | 0.012 | 0.011 | 0.017 | 0.011 |
| TotalTime(↓) | 0.698 | **0.081** | 15.125 | 2.114 | 0.878 |
| StepTime(↓) | **0.003** | 0.008 | 0.076 | 0.106 | 0.004 |

**(c)** Walker2d-Simple

| Metric | Flow | PolyFlow | SafeFlow | RoSD | GaugeFlow |
|---|---|---|---|---|---|
| Safety(↑) | 0.985 | **1.000** | **1.000** | **1.000** | **1.000** |
| MMD(↓) | 9.72-4 | **8.54-4** | 9.72-4 | 1.47-3 | 1.05-3 |
| W(↓) | 3.638 | **3.451** | 3.638 | 3.781 | 3.650 |
| KL(↓) | 2.700 | **2.291** | 2.698 | 3.154 | 2.808 |
| Acc(↓) | **0.034** | 0.035 | **0.034** | 0.036 | **0.034** |
| TotalTime(↓) | 0.403 | **0.081** | 0.598 | 2.043 | 0.649 |
| StepTime(↓) | **0.004** | 0.008 | 0.006 | 0.102 | 0.006 |

**(d)** Walker2d-Hard

| Metric | Flow | PolyFlow | SafeFlow | RoSD | GaugeFlow |
|---|---|---|---|---|---|
| Safety(↑) | 0.705 | **1.000** | **1.000** | **1.000** | **1.000** |
| MMD(↓) | 9.72-4 | **8.75-4** | 9.73-4 | 1.49-3 | 1.02-3 |
| W(↓) | 3.638 | **3.440** | 3.637 | 3.785 | 3.647 |
| KL(↓) | 2.700 | **2.235** | 2.686 | 3.159 | 2.788 |
| Acc(↓) | **0.034** | 0.035 | **0.034** | 0.036 | 0.035 |
| TotalTime(↓) | 0.348 | **0.079** | 4.530 | 1.339 | 0.435 |
| StepTime(↓) | **0.003** | 0.008 | 0.045 | 0.067 | 0.004 |

**(e)** Halfcheetah

| Metric | Flow | PolyFlow | SafeFlow | RoSD | GaugeFlow |
|---|---|---|---|---|---|
| Safety(↑) | 0.000 | **1.000** | **1.000** | **1.000** | **1.000** |
| MMD(↓) | **3.10-3** | 3.60-2 | 1.60-2 | 1.44-2 | 2.78-2 |
| W(↓) | **0.665** | 0.857 | 0.769 | 0.772 | 0.816 |
| KL(↓) | **0.922** | 2.643 | 2.962 | 3.219 | 2.589 |
| Acc(↓) | 3.234 | **2.708** | 2.939 | 3.011 | 2.759 |
| TotalTime(↓) | 0.358 | **0.089** | 10.071 | 2.334 | 0.502 |
| StepTime(↓) | **0.004** | 0.009 | 0.101 | 0.117 | 0.005 |

*Table 9.* Rollout metrics across five different environments. We report the mean and standard deviation over 10 episodes. *Max V. Mag.* denotes the maximum constraint violation magnitude ratio, and *Max V. Dur.* represents the maximum proportion of time steps where constraints are violated. (**Note**: In the Hopper and Walker2d tasks, constraints are enforced solely on the states of the generated window trajectories, without explicitly restricting the executed actions. Consequently, constraint violations may occur during rollouts. However the imposed constraints provide guidance for the rollout trajectories. In contrast, in the HalfCheetah task, constraints are imposed on the action space, thereby ensuring constraint satisfaction during rollouts.)

| Environment | Metric | Flow | PolyFlow (Ours) | SafeFlow | RoSD | GaugeFlow |
|---|---|---|---|---|---|---|
| **Hopper-Simple** | Max V. Mag. ($\downarrow$) | 0.028 | 0.026 | 0.033 | **0.003** | 0.024 |
| | Max V. Dur. ($\downarrow$) | 0.086 | 0.061 | 0.065 | **0.011** | 0.079 |
| | Rollout Return ($\uparrow$) | $2450 \pm 878$ | $\mathbf{3187 \pm 753}$ | $2628 \pm 887$ | $961 \pm 613$ | $2473 \pm 951$ |
| **Hopper-Hard** | Max V. Mag. ($\downarrow$) | 0.456 | **0.137** | 0.325 | 0.256 | 0.228 |
| | Max V. Dur. ($\downarrow$) | 0.350 | 0.330 | 0.319 | **0.224** | 0.327 |
| | Rollout Return ($\uparrow$) | $2450 \pm 878$ | $\mathbf{2949 \pm 838}$ | $2397 \pm 877$ | $937 \pm 539$ | $2851 \pm 944$ |
| **Walker2d-Simple** | Max V. Mag. ($\downarrow$) | **0** | **0** | 0.008 | 0.011 | 0.001 |
| | Max V. Dur. ($\downarrow$) | **0** | **0** | 0.012 | 0.009 | 0.002 |
| | Rollout Return ($\uparrow$) | $5895 \pm 113$ | $\mathbf{6031 \pm 89}$ | $5936 \pm 96$ | $5816 \pm 224$ | $5974 \pm 72$ |
| **Walker2d-Hard** | Max V. Mag. ($\downarrow$) | 0.534 | **0.484** | 0.513 | 1.393 | 0.892 |
| | Max V. Dur. ($\downarrow$) | 0.023 | 0.043 | **0.020** | 0.069 | 0.044 |
| | Rollout Return ($\uparrow$) | $5895 \pm 113$ | $\mathbf{5981 \pm 114}$ | $5961 \pm 95$ | $5716 \pm 577$ | $5783 \pm 618$ |
| **Halfcheetah** | Max V. Mag. ($\downarrow$) | 1.815 | **0** | **0** | **0** | **0** |
| | Max V. Dur. ($\downarrow$) | 0.689 | **0** | **0** | **0** | **0** |
| | Rollout Return ($\uparrow$) | $724 \pm 375$ | $\mathbf{2977 \pm 785}$ | $2034 \pm 892$ | $2083 \pm 580$ | $1399 \pm 309$ |

# H. Safe Locomotion in Isaac Gym Environment

## H.1. Specifications

To further evaluate the efficacy of PolyFlow under high-dimensional dynamics and complex safety constraints, we extend our experiments to quadrupedal locomotion in Isaac Gym environment (Makoviychuk et al., 2021). We use the Unitree Go2 robot, which has 12 actuated joints with 3 degrees of freedom (DoF) per leg (hip, thigh, and calf).

Ensuring physically feasible contact is critical for stable locomotion. In this work, we define safety through friction cone constraints to prevent foot slippage. Given a contact force $\mathbf{f} = [f_x, f_y, f_z]^T$ and friction coefficient $\mu$, the nonlinear friction cone is given by:

$$\sqrt{f_x^2 + f_y^2} \leq \mu f_z, \quad 0 \leq f_z \leq f_{z,max}. \tag{50}$$

In practical implementation, this nonlinear friction cone is commonly linearized into a pyramidal rectangular approximation:

$$|f_x| \leq \mu f_z, \quad |f_y| \leq \mu f_z, 0 \leq f_z \leq f_{z,max}. \tag{51}$$

These constraints can be compactly represented as a set of linear inequalities in the contact force space, expressed as $\mathbf{A}_f \mathbf{f} \leq \mathbf{b}_f$, where:

$$\mathbf{A}_f = \begin{bmatrix} 1 & 0 & -\mu \\ -1 & 0 & -\mu \\ 0 & 1 & -\mu \\ 0 & -1 & -\mu \\ 0 & 0 & 1 \end{bmatrix}, \quad \mathbf{b} = \begin{bmatrix} 0 \\ 0 \\ 0 \\ 0 \\ f_{z,max} \end{bmatrix}. \tag{52}$$

### H.1.1. CONVERT CONSTRAINTS TO THE ACTION SPACE

To enforce safety constraints at the actuation level, we map the contact force constraints into the action space. Consider the robot dynamic equation:

$$\mathbf{M}(\mathbf{q})\ddot{\mathbf{q}} + \mathbf{c}(\mathbf{q}, \dot{\mathbf{q}}) + \mathbf{g}(\mathbf{q}) = \mathbf{B}\tau + \mathbf{J}_{all}^T(\mathbf{q})\mathbf{f}_{all}, \tag{53}$$

where $\mathbf{q} \in SE(3) \times \mathbb{R}^{12}$ denotes the position and quaternion of the floating-base, as well as the joint positions; $\dot{\mathbf{q}} \in \mathbb{R}^{6+12}$ denotes the linear, angular velocity of the base, as well as the joint velocities; $\mathbf{f}_{all}$ denotes the concatenate of all contact Ground Reaction Forces (GRFs) and wrenches; $\tau \in \mathbb{R}^{12}$ denotes joint torques; $\mathbf{J}_{all}^T \in \mathbb{R}^{(6+12) \times 6n_c}$ denotes Jacobians ($n_c$ is the contact leg number), $\mathbf{M} \in \mathbb{R}^{18 \times 18}$, $\mathbf{c} \in \mathbb{R}^{18}$, and $\mathbf{g} \in \mathbb{R}^{18}$ refers to mass matrix, Centrifugal and Coriolis terms, and gravity vector, respectively.

Given that the full Jacobian $\mathbf{J}_{all}^T$ exhibits a block-diagonal structure, we can decouple Equation 53 to isolate the dynamics of individual contacting legs. For a single leg, the relationship between the contact force and joint torque is expressed as:

$$\mathbf{f} = \mathbf{J}^{-T}(\mathbf{h} - \tau), \tag{54}$$

where $\mathbf{J}^{-T} \in \mathbb{R}^{3 \times 3}$ is the inverse transpose of the leg Jacobian, $\mathbf{h} \in \mathbb{R}^3$ represents the aggregated nonlinear dynamic terms, and $\tau \in \mathbb{R}^3$ denotes the leg joint torques. We focus exclusively on the ground reaction force constraints, neglecting moment constraints; consequently, we utilize the translational Jacobian associated with the contact force. Notably, due to the specific 3-DoF configuration of the Unitree Go2 legs, the Jacobian $\mathbf{J}$ is square and full-rank (except at singular configurations), allowing for direct matrix inversion.

Leveraging the invertibility of the Jacobian, the mapping from the contact force space to the joint torque space is convexity-preserving. To further map this to action space, we consider the PD controller formulation:

$$\tau = K_p(a + q_n - q_{cur}) - K_d \dot{q}_{cur}, \tag{55}$$

where $K_p, K_d \in \mathbb{R}^{3 \times 3}$ are diagonal matrices representing the proportional and derivative gains, respectively; $q_n$ denotes the nominal joint positions; and $q_{cur}, \dot{q}_{cur}$ represent the current joint positions and velocities.

By substituting the torque formulation into the force constraints, we can directly derive the safety polytope in the action space via an affine transformation. This yields a set of time-varying linear inequalities $\mathbf{A}a \leq \mathbf{b}$, defined as:

$$\mathbf{A}a \leq \mathbf{b}$$
$$\mathbf{A} = -\mathbf{A}_f \mathbf{J}^{-T} K_p \tag{56}$$
$$\mathbf{b} = \mathbf{b}_f + \mathbf{A}_f \mathbf{J}^{-T}(K_p(q_n - q_{cur}) - K_d \dot{q}_{cur}).$$

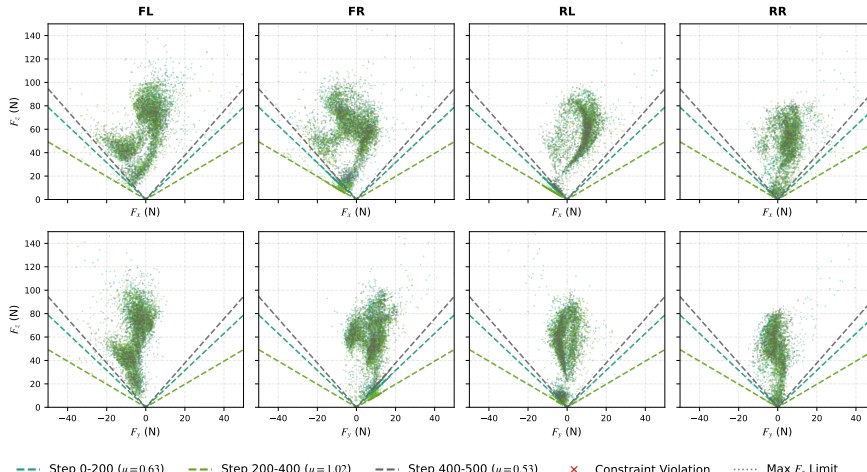

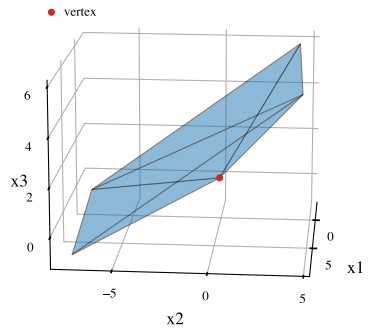

*Figure 11.* Visualization of the feasible action polytope for the Unitree Go2 robot. The region represents the set of safe actions that satisfy the linearized friction cone constraints given the current robot state $q_t$ and $\dot{q}_t$.

*Figure 10.* Distribution of ground reaction forces for stance legs generated by PolyFlow. During rollout, the friction coefficient $\mu$ is randomized to evaluate the policy adherence under varying safety constraints. Dashed lines indicate the boundaries of the friction cone, demonstrating strict safety satisfaction when constraints shift dynamically.

These linear constraints are then integrated into the PolyFlow framework to ensure feasible and safe locomotion commands.

It is worth noting that the constraint parameters $\mathbf{A}(q_t)$ and $\mathbf{b}(q_t, \dot{q}_t)$ are state-dependent and evolve dynamically with the robot's instantaneous configuration. Consequently, the safety constraints form a time-varying polytope in the action space. Figure 11 illustrates the geometry of this feasible action region at a single timestep.

### H.2. Training Details

Given the state-dependent nature of the safety constraints (as defined by $\mathbf{A}(q)$ and $\mathbf{b}(q, \dot{q})$), accurate prediction of future constraint matrices is infeasible. Consequently, we restrict the prediction horizon of the flow model to a single time step ($H = 1$). In this setup, the model maps the current observation directly to the immediate action. To mitigate covariate shift and reduce compounding errors during rollout, we employ the DAgger (Dataset Aggregation) algorithm (Ross et al., 2011) to train the PolyFlow model.

**Expert Policy Training.** We initially train an expert policy using PPO (Schulman et al., 2017) algorithm. The policy architecture and hyperparameters used for data generation are adapted from (Rudin et al., 2022). To enhance robustness, domain randomization is applied to the ground friction coefficient, motor strength, base mass, and external disturbance forces. Furthermore, we incorporate an auxiliary penalty term into the reward function to discourage constraint violations, imposing a "soft constraint" on the expert's behavior. It is important to note that the trajectories generated by this expert are not strictly safe and may still exhibit occasional constraint violations.

**PolyFlow Training.** Leveraging the pre-trained expert, we train the PolyFlow model within the Isaac Gym environment using the DAgger framework. The specific hyperparameters used for training are detailed in Table 10.

*Table 10.* Hyperparameters for PolyFlow training in Go2 locomotion task.

| Parameter | Value |
| --- | --- |
| Optimizer | Adam |
| Learning Rate | $1 \times 10^{-4}$ |
| Batch Size | 512 |
| DAgger Iterations | 100 |
| Samples per Iteration | $4096 \times 50$ |
| Training Epochs per Iteration | 15 |
| $\beta$ Decay | 0.98 |

| Parameter | Value |
| --- | --- |
| Backbone Network | DiT |
| Hidden Dim. | 256 |
| Trajectory Encoder Layers | 4 |
| Constraint Encoder Layers | 4 |
| Discrete Steps | 10 |

# I. Ablation Study

## I.1. Necessity of Constraint Embedding

To clarify the role of the constraint encoding architecture, we conducted an ablation study across three tasks. Since the set of linear inequalities is inherently unordered, any effective encoding must be permutation-invariant. We compared three configurations:

- **PolyFlow-attn**: Our default architecture, where constraint embeddings are fused into the latent space via cross-attention.

- **PolyFlow-mlp**: Constraints are processed by an MLP, aggregated via a sum operator (to ensure permutation-invariance), and concatenated with trajectory features.

- **PolyFlow-w/o embed**: In this case, no constraint information is provided. The network relies solely on state inputs.

As shown in Table 11 and Table 12, for tasks with static constraints (Maze2d, HalfCheetah), the "w/o" baseline performs surprisingly well. We hypothesize that the model implicitly learns the fixed boundaries from the state distribution. However, in the Unitree Go2 task, where constraints are state-dependent and time-varying, explicit encoding becomes essential for the model to generalize to shifting safe boundaries.

*Table 11.* Performance Comparison on Maze2d and HalfCheetah with different constraint encoding architectures.

| Task | Architecture | **MMD** ($\downarrow$) | **W** ($\downarrow$) | **KL** ($\downarrow$) | **Acc** ($\downarrow$) | **TotalTime(s)** ($\downarrow$) |
|---|---|---|---|---|---|---|
| Maze2d | PolyFlow-attn | 6.20e-6 | 0.041 | 0.134 | 6.15e-3 | 2.11 |
| | PolyFlow-mlp | 2.62e-6 | 0.033 | 0.050 | 6.37e-3 | 0.58 |
| | PolyFlow-w/o embed | 1.91e-6 | 0.033 | 0.066 | 6.03e-3 | 0.50 |
| HalfCheetah | PolyFlow-attn | 3.60e-2 | 0.857 | 2.643 | 2.708 | 0.089 |
| | PolyFlow-mlp | 2.98e-2 | 0.835 | 2.451 | 2.735 | 0.061 |
| | PolyFlow-w/o embed | 3.80e-2 | 0.836 | 2.471 | 2.717 | 0.060 |

*Table 12.* Rollout Performance on Unitree Go2 (Dynamic Constraints).

| Architecture | **Episode Length** ($\uparrow$) | **Tracking Error** ($\downarrow$) |
|---|---|---|
| PolyFlow-attn | $305 \pm 37$ | $0.026 \pm 0.007$ |
| PolyFlow-mlp | $311 \pm 64$ | $0.026 \pm 0.007$ |
| PolyFLow-w/o embed | $227 \pm 15$ | $0.046 \pm 0.008$ |

## I.2. Coupling between Weight Predictor and Direction Predictor

Considering the conceptual question that absolute displacement is inherently direction-dependent, we conducted an ablation comparing our decoupled architecture against a "coupled" variant (where the weight predictor $\gamma_\theta$ receives direction info from $d_\theta$).

The full ablation results are presented in Table 13. Decoupling the step size from the direction has only a marginal effect on overall performance. This suggests that the step-size predictor can implicitly infer directional context from the shared latent features (state and constraint embeddings) without explicit direction input.

## I.3. Ray Shooting Operators

We evaluate the influence of different ray shooting operations on the overall model performance. Specifically, we compare three variants of the minimization operator defined in Eq. 40, each representing a different trade-off between smoothness and precision:

- **Hard:** The standard minimization operator, $\alpha_{\text{hard}} = \min\{t_1, t_2, \ldots, t_n\}$, where $t_i$ denotes the distance to the $i$-th boundary intersection.

*Table 13.* Coupled vs. Decoupled Architecture Performance.

| Task | Architecture | MMD (↓) | W (↓) | KL (↓) | Acc (↓) |
|------|-------------|---------|-------|--------|---------|
| Hopper-Simple | Coupled | 1.74e-4 | 1.114 | 0.393 | 0.011 |
|  | Decoupled | 2.37e-4 | 1.098 | 0.367 | 0.012 |
| Walker2d-Simple | Coupled | 8.54e-4 | 3.451 | 2.291 | 0.035 |
|  | Decoupled | 9.69e-4 | 3.564 | 2.344 | 0.036 |
| HalfCheetah | Coupled | 3.60e-2 | 0.857 | 2.643 | 2.708 |
|  | Decoupled | 3.10e-2 | 0.841 | 2.458 | 2.727 |
| Maze2D | Coupled | 6.20e-6 | 0.041 | 0.134 | 0.098 |
|  | Decoupled | 8.23e-6 | 0.038 | 0.105 | 0.098 |

- **Softmin:** A smooth approximation using the LogSumExp operator, defined as $\alpha_{\text{softmin}} = -\frac{1}{\beta} \ln \left( \sum_{i=1}^{n} e^{-\beta t_i} \right)$.

- **Boltzmann:** A probabilistic weighted average, given by $\alpha_{\text{boltzmann}} = \sum_{i=1}^{m} w_i \cdot t_i = \frac{\sum_{i=1}^{n} t_i e^{-\beta t_i}}{\sum_{j=1}^{n} e^{-\beta t_j}}$.

Theoretically, the value derived from the **Softmin** is strictly lower than the true minimum (**Hard**), making it a conservative approximation that satisfies safety constraints. Conversely, **Boltzmann** yields an expected minimum, which does not provide strict theoretical guarantees for constraint satisfaction.

We evaluated these variants on the HalfCheetah task, with results summarized in Table 14. While all variants exhibit comparable performance in terms of distribution matching, the **Hard** variant achieves the highest rollout returns. This performance gap stems from the fact that optimal actions in HalfCheetah often reside on the constraint boundaries (see Figure 9). Consequently, the precise boundary estimation provided by the **Hard** operation captures these edge cases more effectively than smoothed approximations. Furthermore, as expected, the **Boltzmann** variant fails to ensure strict safety, resulting in minor safety violations. Based on these findings, we adopt the **Hard** operator as the default configuration for PolyFlow.

### I.4. OT-Based Batch Coupling

To minimize the approximation error identified in the second term on the RHS of Theorem 4.4, we incorporate OT coupling within training batches to reduce trajectory crossings. We evaluate the effectiveness of this approach on the HalfCheetah task. As reported in Table 14, the OT batch coupling method achieves distribution matching performance comparable to the baseline while yielding slightly higher returns during rollouts.

*Table 14.* Performance comparison on the HalfCheetah task with different configurations. We compare the standard RS (Hard) with the Softmin and Boltzmann variants ($\beta = 80$). Hard+OT denotes using optimal transport to pair samples within the training batch.

| Metric | Hard | Softmin($\beta = 80$) | Boltzmann($\beta = 80$) | Hard+OT |
|--------|------|----------------------|------------------------|---------|
| R (↑) | 1.000 | 1.000 | 0.705 | 1.000 |
| MMD (↓) | 0.036 | **0.035** | **0.035** | 0.036 |
| W (↓) | 0.857 | **0.854** | 0.855 | 0.858 |
| KL (↓) | 2.643 | 2.680 | 2.655 | **2.488** |
| Acc (↓) | **2.708** | 2.723 | 2.716 | 2.729 |
| TotalTime (s) (↓) | 0.089 | 0.095 | 0.104 | **0.074** |
| StepTime (s) (↓) | 0.009 | 0.010 | 0.010 | **0.007** |
| Max V. Mag. (↓) | 0.000 | 0.000 | 0.003 | 0.000 |
| Max V. Dur. (↓) | 0.000 | 0.000 | 0.396 | 0.000 |
| Rollout Return (↑) | $2977 \pm 785$ | $2773 \pm 543$ | $2729 \pm 565$ | $\mathbf{3026 \pm 301}$ |

### I.5. Integration Steps

We examine the performance of PolyFlow across varying integration steps $N \in \{2, 5, 10, 20\}$. As illustrated in Figure 12, PolyFlow demonstrates high efficiency in the low-step regime, achieving its highest rollout returns at $N = 2$ and exhibits

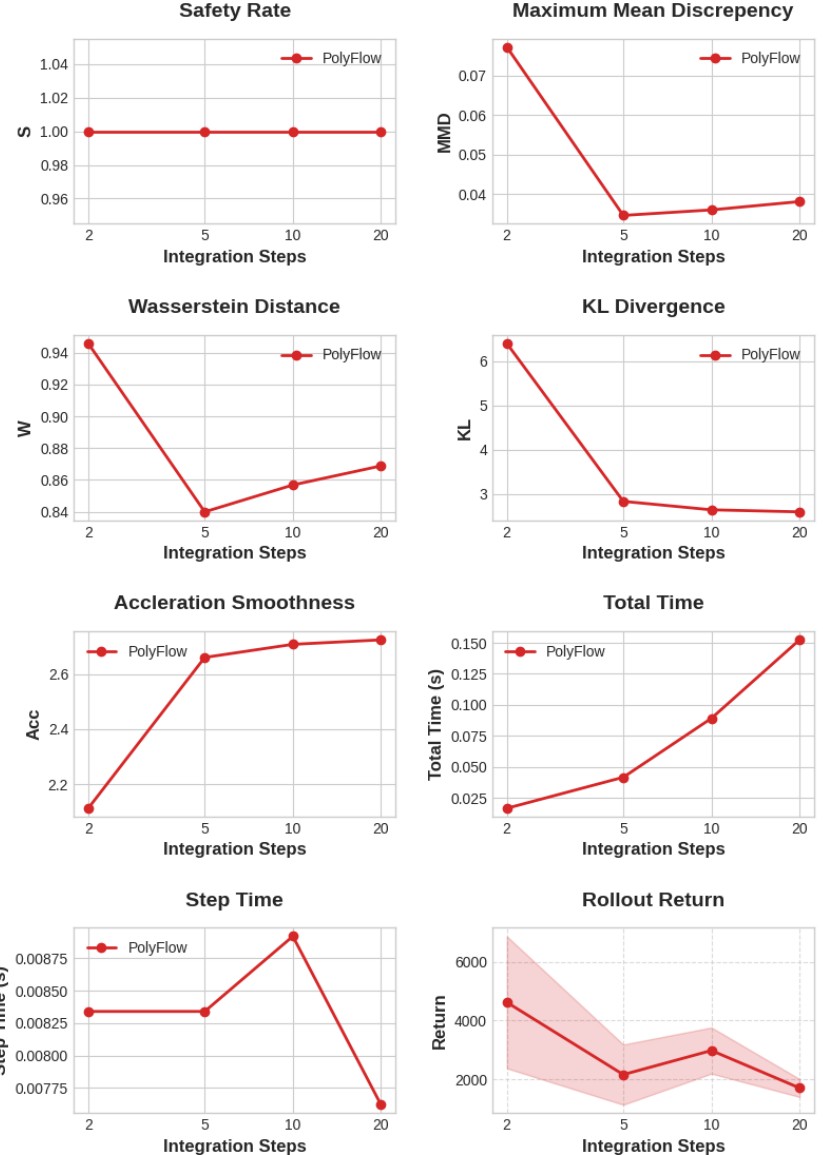

*Figure 12.* Impact of integration steps on PolyFlow performances in HalfCheetah task.

supeior trajectory smoothness. As $N$ increases, the proposed method shows a more precise alignment with the target distribution through finer discretization. Crucially, PolyFlow maintains a perfect Safety Rate ($R = 1$) across all steps, demonstrating exceptional robustness to the choice of integration density.

## J. Why Standard Flow Matching Fails at Constraint Satisfaction

Constraint violations in continuous-time flow models generally stem from an entanglement of several factors: unsafe trajectories present in the training data, inherent model fitting errors, and numerical integration errors during the sampling phase. To empirically isolate these factors and demonstrate why traditional flow matching fails—thereby motivating our discrete-time, explicitly constrained formulation—we conducted controlled experiments on the Maze2D task.

**The Insufficiency of Data Filtering.** First, we investigated whether training exclusively on perfectly clean data could eliminate constraint violations. We filtered the original training dataset to remove all unsafe trajectories and trained a new unconstrained model, denoted as **FlowFiltered**.

As shown in Table 15, even when trained exclusively on safe data, the FlowFiltered model fails to achieve strict

safety (Safety Rate $= 0.965$). The remaining violations are primarily caused by inherent neural network fitting errors and discretization errors during generation. This confirms that data filtering alone is insufficient to guarantee safety, underscoring the fundamental necessity of PolyFlow's explicit constraint embedding.

*Table 15.* Performance comparison of models trained on standard vs. strictly safe data (evaluated at $N = 10$ steps).

| Metric | Flow ($N = 10$) | FlowFiltered ($N = 10$) | PolyFlow ($N = 10$) |
|---|---|---|---|
| Safety ($\uparrow$) | 0.845 | 0.965 | **1.000** |
| MMD ($\downarrow$) | 3.50e-5 | 4.33e-5 | **6.20e-6** |
| W ($\downarrow$) | 5.41e-2 | 5.55e-2 | **4.08e-2** |
| KL ($\downarrow$) | 2.66e-1 | 2.30e-1 | **1.34e-1** |

**The Vulnerability of Numerical Integration.** Second, to empirically isolate the impact of numerical integration errors (discretization), we evaluated an optimized continuous flow model. To eliminate data artifacts and minimize fitting errors, we utilized the aforementioned strictly safe dataset and extended the training to $10^6$ steps to ensure optimal model convergence. We then evaluated this continuous flow using a 4th-order Runge-Kutta (RK4) solver across various integration steps.

*Table 16.* Continuous flow performance using an RK4 solver across varying integration steps on the Maze2D task.

| Metric | MMD ($\downarrow$) | W ($\downarrow$) | KL ($\downarrow$) | Safety Rate ($\uparrow$) | Cur ($\downarrow$) | Acc ($\downarrow$) |
|---|---|---|---|---|---|---|
| Flow1-rk4 | 8.00e-5 | 2.82e-1 | 1.370 | 0.000 | 2.262 | 1.800 |
| Flow10-rk4 | 1.68e-5 | 3.87e-2 | 1.15e-1 | 0.755 | 1.342 | 8.65e-2 |
| Flow100-rk4 | 2.42e-5 | 4.03e-2 | 9.71e-2 | 0.945 | 0.098 | 5.99e-3 |
| Flow200-rk4 | 2.47e-5 | 4.03e-2 | 9.69e-2 | 0.945 | 0.067 | 4.10e-3 |
| Flow500-rk4 | 2.50e-5 | 4.04e-2 | 9.70e-2 | 0.945 | 0.054 | 3.40e-3 |
| Flow200-rk4-jump0.9 | 2.04e-5 | 3.97e-2 | 1.11e-1 | 0.790 | 1.246 | 7.75e-2 |
| PolyFlow10 | **6.20e-6** | 4.08e-2 | 1.34e-1 | **1.000** | 0.098 | 6.15e-3 |

The results in Table 16 illustrate that even when employing a higher-order ODE solver on a well-fitted model, reducing integration steps systematically degrades both generation quality and the safety rate. Notably, even with 500 integration steps, the continuous model plateaus at a safety rate of $0.945$, failing to guarantee strict constraint satisfaction.

Crucially, we conducted a targeted test, denoted as `Flow200-rk4-jump0.9`, where fine integration steps are taken until $t = 0.9$, followed by a single linear jump to $t = 1.0$. This single large step caused the safety rate to plummet from $0.945$ to $0.790$. This highlights a critical vulnerability: when the generation path approaches a target distribution located near a constraint boundary, large integration steps risk overshooting into unsafe regions.

**Advantages of the Discrete-Time Formulation.** Unlike continuous flows that rely on external solvers to approximate an ODE, PolyFlow is mathematically discrete by definition. By embedding the constraints into the architecture and ensuring that every discrete update explicitly resides within the feasible region via ray-shooting, PolyFlow fundamentally eliminates numerical integration overshoot. This paradigm allows for strict safety satisfaction even with very few inference steps ($N = 10$), entirely bypassing the need for expensive iterative projection solvers.

## K. Can PolyFlow be Extended to Equality Constraints?

PolyFlow can be extended to handle linear equality constraints ($Cx = u$) alongside inequality constraints ($Ax \leq b$) using two primary strategies:

**1. Null Space Projection Operator:** When equality constraints are introduced, the predicted direction vector $d_\theta$ must strictly lie within the null space of $C$. This means $C(x_0 + \lambda d_\theta) = u + \lambda C d_\theta = u$, implying $C d_\theta = 0$. We can append a differentiable projection operator following the direction prediction network:

$$d_{out} = (I - C^T(CC^T)^{-1}C)d_\theta \tag{57}$$

This operation guarantees $C d_{out} = 0$. By utilizing $d_{out}$ for the ray-shooting computations, the model inherently satisfies

both sets of constraints. For $m$ equality constraints and state dimension $n$, the added complexity is $\mathcal{O}(m^2 n + m^3)$. Since $m \ll n$, this projection will not significantly impact latency.

**2. Dimensionality Reduction:** For static $Cx = u$, we can precompute a particular solution $x_e$ and a null-space basis matrix $N$. Valid solutions are reparameterized as $x = x_e + Ny$ (where $y$ is a free vector with lower dimensionality). Substituting this into the inequality constraints yields:

$$A(x_e + Ny) \leq b \implies ANy \leq b - Ax_e \tag{58}$$

By defining $A_y = AN$ and $b_y = b - Ax_e$, we can train PolyFlow entirely in the lower-dimensional $y$-space and map back via $x = x_e + Ny$ during inference.

