# OpenReview forum: "PolyFlow: Safe and Efficient Polytope-Constrained Flow Matching with Constraint Embedding and Projection-free Update"
_ICML.cc/2026/Conference — ICML 2026 regular_

### Official Review · Reviewer_DQkG · 2026-03-11

**Soundness:** 3
**Presentation:** 2
**Significance:** 3
**Originality:** 3
**Overall Recommendation:** 5
**Confidence:** 3

**Summary:**

This paper proposes a polytope-constraint flow matching framework. The method employs a ray-shooting operator to restrict the vector field within a safety polytope and introduces a novel parameterization architecture.

**Compliance With Llm Reviewing Policy:**

Affirmed.

**Final Justification:**

My concerns are addressed and I have raised my score.

**Key Questions For Authors:**

1. What is the applicability of the linear safety polytope? Which real-world constraints can be modeled as such, and which cannot?
2. Will code and/or data be made available to ensure reproducibility?

**Limitations:**

yes

**Strengths And Weaknesses:**

**Strengths**
- The proposed approach is novel and computationally efficient for handling polytope constraints, achieving zero constraint violation.
- It outperforms several recent strong baselines across various generative quality metrics.
- Real-robot experiments on the Unitree Go2 demonstrate its practical applicability.

**Weaknesses**
- The applicability of the safety polytope to real-world problems (e.g., robot control) is not sufficiently discussed. It remains unclear which types of constraints can be modeled as a safety polytope and which cannot.
- The reproducibility of the work is not addressed (e.g., no mention of code or data release).
- Minor issues include
    - Missing references at lines 865 and 972 (indicated by “(?)”).
    - Incorrect quotation marks in several places.
    - Inconsistent notation: “Wasserstein-2” vs. “2-Wasserstein” should be unified.

---

> ### Author Rebuttal · Authors · 2026-03-30
>
> We sincerely thank you for your encouraging and constructive review. Your recognition of our method's novelty, efficiency, and real-world applicability is highly appreciated. Below, we address your questions and outline the improvements we will make to the manuscript.
>
> ## 1. Applicability of the Linear Safety Polytope
>
> We appreciate this valuable question regarding the practical scope of linear safety polytopes. In fact, **any bounded linear constraints can be formulated as a convex polytope**. Robotics provides excellent examples of the broad applicability of convex polytope constraints:
>
> - **Kinematic Constraints:** Joint position bounds, velocity limits, and workspace boundaries for end-effector trajectories are inherently bounded linear inequalities, fitting perfectly within the polytope definition.
> - **Dynamic and Actuation Constraints:** Physical limitations such as motor torque saturation and the maximum allowed wrench of an end-effector naturally translate into linear polytope constraints.
> - **Task and Environmental Constraints:** Interaction limits, such as feasible wrench spaces at contact points or friction constraints for legged robots to prevent slipping, can be effectively modeled as convex polytopes.
>
> For real-world safety limits that are not natively linear or convex, standard engineering workarounds are widely used and fully compatible with PolyFlow:
>
> - **Non-Convex Spaces:** Obstacle avoidance in trajectory planning often involves non-convex free space. By applying established convex decomposition methods, we can break the global non-convex space into local convex polytopes, allowing our method to apply seamlessly (as demonstrated in our Maze2D experiments).
> - **Non-Linear Boundaries:** Certain dynamic boundaries, such as rotation limits in SO(3) space or exact second-order cone friction models, cannot be directly encoded as polytopes A standard practice is to address these open problems using linear approximation techniques. For instance, a non-linear friction cone is commonly linearized into a pyramidal rectangular approximation. This approximation allows us to model the boundary as a convex polytope, which we have detailed in Appendix H.1.
>
>
> ## 2. Reproducibility, Code, and Data
>
> We sincerely apologize for omitting the code repository link in the manuscript. Actually, **we have uploaded our complete project code in the supplementary materials.** The submitted zip file contains the full training and inference pipelines for both the Maze2D and Gym experiments, as well as the Maze2D training dataset. Although the Gym datasets are not bundled in the file, they can be easily downloaded using the standard Minari API. Upon acceptance, we will release a polished, well-documented version of our codebase, datasets, and pre-trained models on GitHub to ensure full reproducibility for the broader community.
>
> ## 3. Missing References, Typos, and Notation
>
> We appreciate your careful reading and attention to detail.
>
> - We have identified the missing citations at lines 865 and 972 and will add the correct references in the final version.
> - We will fix all incorrect quotation marks.
> - We will strictly unify the notation to "2-Wasserstein" throughout the entire manuscript to ensure absolute consistency.
>
>
> We sincerely hope these responses address your concerns. Should you have any remaining questions or require further details, we are more than happy to engage in further discussion.

---

> > ### Author Rebuttal · Reviewer_DQkG · 2026-04-03
> >
> > Thank you for your response. My concerns are addressed. I believe this is a good paper and the issues are easy to address by revision. I will raise my score.

---

> > > ### Author Response · Authors · 2026-04-03
> > >
> > > We are very grateful for the reviewer's encouraging comments and the decision to raise the score.
> > > As we noticed that the score hasn't been updated in the portal yet, we would like to kindly check if there are any remaining steps or further clarifications needed from our side to help finalize the evaluation. Thank you again for your support and for helping us improve our paper. Best regards.

---

### Official Review · Reviewer_j5nG · 2026-03-12

**Soundness:** 4
**Presentation:** 2
**Significance:** 3
**Originality:** 3
**Overall Recommendation:** 5
**Confidence:** 3

**Summary:**

PolyFlow is an interesting approach to restrict flow-based generative models (in discrete time) to polytopes. Similar to how a Hit-and-Run sampler works, we first choose a direction, then shoot a ray and record the intersection with the constraints (although the paper appears to be more motivated by frank-wolf). Along this ray we then jump towards the next point. It seems like a straightfoward, scalable way define a flow on a polytope, as a hit-and-run sampler is an also a scalable way to define a MCMC for uniformal polytope sampling. The paper then follows up with an experimental section that features challenging real-world applications.

**Compliance With Llm Reviewing Policy:**

Affirmed.

**Final Justification:**

The main issue when reviewing the paper was question 1, which was properly addressed in the rebuttal. The paper tackles a significant problem (polytope constrained generation), with a good experimental section to support soundness. The idea appears original and straightforward to apply when needed, which is an also important. If the question 1 is properly addressed then the presentation is good. Therefore I recommend acceptance.

**Key Questions For Authors:**

Q1: In the algorithm 1, it seems like the step network does not get the step-length or even direction. Is the step size really independent of direction d? If so, I think this choice should be highlighted. As the step-size as the output is really relative to the available length along the ray towards the boundary, this would make it hard for the network to output step-size in absolute terms in the "data-space".

Q2: Can you think of a way to include equality constraints in addition to inequality constraints to the algorithm?

Q3: Could this method be extended to handle continous time flows?

**Limitations:**

yes

**Strengths And Weaknesses:**

The paper is sound: Ray shooting is an established technique and combining it with (discrete time) flow models is straightforward. The key idea here is to disentangle direction and step length and then apply ray-shooting ideas. The simplicity is an advantage and supports the soundness from the theoretical side. Experimentally, the paper is supported by multiple real-world applications demonstrating feasibility of the approach, also on commonly used problems like the Gym-tasks.

The presentation is good: The paper is clearly written and understandable, although a few questions remain. The overall narrative is straightforward to understand and the experiments are detailed.

The problem is significant: Polytope constrained problems are significant and have real-word applications, even more so if the polytopes are not static but are allowed to vary (as demonstrated in the maze-experiment).

The paper is original: The work is original, although new insights into the problem are more limited, which is fine. The paper combines established ideas, which in itself is valuable.

---

> ### Author Rebuttal · Authors · 2026-03-30
>
> We sincerely thank you for evaluating our work as sound, clear, and significant. Your insightful questions helped refine our architectural justifications and theoretical boundaries. Below, we address your comments.
>
> ## Q1. Step-Size and Direction Coupling
>
> We apologize for the confusion regarding Algorithm 1 and Figure 2, which were simplified for visual brevity. In our actual implementation, the weight predictor $\gamma_{\theta}$ _does_ receive the output features of the direction predictor $d_{\theta}$. We will correct the manuscript to explicitly show that $\gamma_{\theta}$ is conditioned on $d_{\theta}$.
>
> Regarding your excellent conceptual question—that absolute displacement is inherently direction-dependent—we conducted an ablation comparing our coupled architecture against a "decoupled" variant (where $\gamma_{\theta}$ receives no direction info).
>
> **Table: Coupled vs. Decoupled Architecture Performance**
>
> | **Task**        |           | MMD     | W     | KL    | Acc   |
> | - | - | - | - | - | - |
> | Hopper-Simple   | Coupled   | 1.74e-4 | 1.114 | 0.393 | 0.011 |
> |    | Decoupled | 2.37e-4 | 1.098 | 0.367 | 0.012 |
> | Walker2d-Simple | Coupled   | 8.54e-4 | 3.451 | 2.291 | 0.035 |
> |    | Decoupled | 9.69e-4 | 3.564 | 2.344 | 0.036 |
> | HalfCheetah     | Coupled   | 3.60e-2 | 0.857 | 2.643 | 2.708 |
> |                 | Decoupled | 3.10e-2 | 0.841 | 2.458 | 2.727 |
> | Maze2D          | Coupled   | 6.20e-6 | 0.041 | 0.134 | 0.098 |
> |                 | Decoupled | 8.23e-6 | 0.038 | 0.105 | 0.098 |
>
> **Architectural Robustness:** Decoupling the step size from the direction has only a marginal effect on overall performance. This suggests that the step-size predictor can implicitly infer directional context from the shared latent features (state/constraint embeddings) without explicit direction input.
> We will include these full ablation results in the revised Appendix.
>
> ## Q2. Including Equality Constraints
>
> We appreciate this constructive question. PolyFlow can be extended to handle linear equality constraints ($Cx = u$) alongside inequality constraints ($Ax \le b$) using two strategies:
>
> **1. Null Space Projection Operator**
> When equality constraints are introduced, the predicted direction vector $d_\theta$ must strictly lie within the null space of $C$. This means $C(x_0+\lambda d_\theta)=u+\lambda Cd_\theta=u$, implying $Cd_\theta=0$. We can append a differentiable projection operator following the direction prediction network:
> $$d_{out}=(I-C^T(CC^T)^{-1}C)d_\theta$$
> This operation guarantees $Cd_{out}=0$. By utilizing $d_{out}$ for the ray-shooting computations, the model inherently satisfies both sets of constraints. For $m$ equality constraints and state dimension $n$, the added complexity is $\mathcal{O}(m^{2}n+m^{3})$. Since $m \ll n$, this projection will not significantly impact latency.
>
> **2. Dimensionality Reduction**
> For static $Cx = u$ , we can precompute a particular solution $x_{e}$ and null-space basis matrix $N$. Valid solutions are reparameterized as $x=x_{e}+Ny$ (where $y$ is a free vector with lower dimensionality). Substituting this into inequality constraints yields:
> $$A(x_e+Ny) \le b \implies ANy \le b-Ax_e$$
> By defining $A_y=AN$ and $b_y=b-Ax_e$, we can train PolyFlow entirely in the lower-dimensional $y$-space and map back via $x=x_{e}+Ny$ during inference.
>
> ## Q3. Extending to Continuous-Time Flows
>
> Extending PolyFlow to continuous-time flows is theoretically possible but requires bridging a fundamental space mismatch.
>
> Continuous flows output in velocity space ($v_\theta$), generating paths via integration: $x_{i+1}=x_i+v_\theta(x_i, i)\delta t$. Discrete flows output directly in the state space ($\Delta_\theta$). Since safety constraints are defined in the state space, ray-shooting cannot be trivially applied to velocity outputs.
>
> We can, however, bridge this gap by mapping state-space constraints ($h(x) \ge 0$) to velocity-space using Control Barrier Functions (CBFs):
> $$\nabla h(x)^Tv \ge -\alpha(h(x))$$
> where $\alpha(\cdot)$ is a strictly monotonically increasing function with $\alpha(0)=0$. CBF theory guarantees that if the velocity vector strictly satisfies this inequality everywhere, the resulting trajectory maintains forward invariance. By applying PolyFlow's ray-shooting to these transformed linear velocity constraints, the continuous flow could theoretically remain safe.
>
> It is crucial to highlight why we deliberately chose discrete-time flows. Continuous flows invariably rely on numerical integration. Accumulated integration errors introduce a severe risk of constraint violation, effectively breaking theoretical safety guarantees. By adopting a discrete-time formulation, PolyFlow enforces strict boundaries directly in the state space, completely eliminating the numerical integration errors.
>
> We sincerely hope these responses address your concerns. Should you have any remaining questions or require further details, we are more than happy to engage in further discussion.

---

> > ### Author Rebuttal · Reviewer_j5nG · 2026-04-02
> >
> > thank you for addressing my points and concerns. Especially clarifying Q1 helps with understanding the algorithm, which was my biggest question mark after reading the paper. I will raise my score.

---

> > > ### Author Response · Authors · 2026-04-03
> > >
> > > We sincerely thank the reviewer for the positive feedback and for the time spent evaluating our rebuttal.

---

### Official Review · Reviewer_6UyV · 2026-03-12

**Soundness:** 3
**Presentation:** 3
**Significance:** 3
**Originality:** 3
**Overall Recommendation:** 4
**Confidence:** 3

**Summary:**

This paper proposes a new framework for constrained generative modeling in physical applications where strict constraint satisfaction is required. Existing approaches typically enforce safety through projection-based corrections during inference, which can be computationally expensive and may distort the learned flow dynamics. Instead, the authors introduce a ray-shooting formulation that embeds polytope constraints directly into the generative flow during training. By constructing updates along safe line segments inside the feasible region, the method guarantees that each generation step remains within the constraint set without requiring expensive projection operations.

This design reduces interference with the target distribution while improving inference efficiency. Experimental results across several tasks—including a 2D maze and multiple control benchmarks—show that the proposed method maintains perfect or near-perfect safety rates while achieving competitive or improved distribution matching compared to existing constrained flow approaches.

**Compliance With Llm Reviewing Policy:**

Affirmed.

**Final Justification:**

The rebuttal has satisfactorily addressed my concerns, particularly clarifying the motivation behind the discrete-time formulation and providing additional insights into the role of the constraint encoding. These clarifications strengthen my confidence in both the technical soundness and practical relevance of the proposed approach. Given the novelty of the method, its strong empirical performance, and the authors’ thorough response to the raised issues, I am now positive about the paper and support its acceptance.

**Key Questions For Authors:**

- Could the authors provide empirical comparisons using higher-order ODE solvers or smaller step sizes to better illustrate the practical advantage of the proposed discrete formulation?
- The proposed architecture includes a dedicated constraint encoder that conditions the model on the polytope constraints. How sensitive is the method to this design choice? For example, what happens if the constraints are not explicitly encoded or if simpler representations are used?

**Limitations:**

Yes.

**Strengths And Weaknesses:**

Strength:
- The key idea of implicitly constructing a safe vector field by generating trajectories that remain within the feasible set (via line segments inside the polytope) is interesting and conceptually different from prior projection- or correction-based approaches.
- The proposed formulation ensures that each update remains within the feasible region without requiring additional optimization steps at inference time, which is appealing for safety-critical applications.
- By avoiding projection or QP-based correction steps, the approach demonstrates improved inference speed compared to several existing constrained flow methods.
- The paper evaluates the method on multiple settings, including a 2D maze task and five control environments, showing consistent improvements in safety while maintaining good distributional fidelity.

Weakness:

**Motivation for the discrete-time formulation is not fully convincing.**

The paper emphasizes the importance of the discrete-time formulation to avoid numerical integration errors. However, continuous-time ODE models must ultimately be discretized during inference as well. It would be helpful to better clarify whether the discrete formulation provides a practical advantage beyond theoretical rigor. For example, comparisons with higher-order ODE solvers (even if slower) could help evaluate whether safety violations in continuous formulations are primarily due to discretization artifacts.

**Limited ablation on the constraint encoding architecture.**

The proposed framework introduces a dedicated network architecture to encode the constraint set. However, the manuscript does not clearly analyze the contribution of this component. An ablation study evaluating performance without explicit constraint encoding (or using simpler encodings) would help clarify how critical this architectural design is to the overall performance.

Overall, the paper proposes a novel and well-motivated approach to constrained generative modeling by embedding polytope constraints directly into the flow dynamics using a ray-shooting formulation. This design provides safety by construction and avoids expensive projection-based corrections during inference, which is particularly appealing for safety-critical physical systems. The empirical results demonstrate strong safety performance together with competitive distribution matching and improved inference efficiency across several tasks. While some aspects of the method—such as the necessity of the discrete-time formulation and the contribution of the constraint encoding architecture—could benefit from additional clarification or ablation studies, these concerns are relatively minor compared to the overall contribution. Given the conceptual novelty and promising empirical results, I lean toward acceptance.

---

> ### Author Rebuttal · Authors · 2026-03-30
>
> We sincerely thank you for your positive assessment and constructive feedback. Below, we address your questions in detail.
>
> ## Q1. Motivation for the Discrete-Time Formulation & ODE Solver Comparisons
> Constraint violations in continuous-time models result from entangled factors: out-of-distribution training data, model fitting errors , and numerical integration errors during sampling. To empirically isolate the impact of discretization, we conducted a controlled experiment on the Maze2D task:
> 1. Eliminating Data Artifacts: We filtered the dataset to include only strictly safe trajectories.
> 2. Minimizing Fitting Error: We extended training to $10^6$ steps to ensure optimal model convergence.
>
> We then evaluated this optimized continuous flow using a 4th-order Runge-Kutta (**RK4**) solver across various integration steps.
>
> **Table 1: Continuous Flow Performance with RK4 Solver (Maze2D)**
>
> |**Metric**|**MMD**|**W**|**KL**|**SafetyRate**|**Cur**|**Acc**|
> |-|-|-|-|-|-|-|
> |Flow1-rk4|8.00e-5|2.82e-1|1.370|0.000|2.262|1.800|
> |Flow10-rk4| 1.68e-5 |3.87e-2|1.15e-1|0.755|1.342| 8.65e-2|
> |Flow100-rk4| 2.42e-5 | 4.03e-2 | 9.71e-2 | 0.945 | 0.098| 5.99e-3|
> |Flow200-rk4| 2.47e-5 | 4.03e-2 | 9.69e-2 | 0.945 | 0.067| 4.10e-3|
> |Flow500-rk4| 2.50e-5 | 4.04e-2 | 9.70e-2 | 0.945 | 0.054| 3.40e-3|
> |Flow200-rk4-jump0.9 | 2.04e-5 | 3.97e-2 | 1.11e-1 | 0.790 | 1.246| 7.75e-2|
> |PolyFlow10| 6.20e-6 | 4.08e-2 | 1.34e-1 | 1.000 | 0.098| 6.15e-3|
>
>  **Analysis of Numerical Integration Error**:
> Even with a higher-order ODE solver like RK4 on a well-fitted model, reducing integration steps systematically degrades both generation quality and the safety rate. Crucially, the `Flow200-rk4-jump0.9` test—where fine steps are taken until $t=0.9$ followed by a single linear jump to $t=1.0$—caused the safety rate to drop from 0.945 to 0.790. This highlights a critical vulnerability: when the generation path approaches a target distribution near a boundary, large integration steps risk overshooting into unsafe regions.
>
> **Advantages of the Discrete-Time Formulation**:
> Unlike continuous flows that rely on external solvers to approximate an ODE, PolyFlow is mathematically discrete by definition. By ensuring every discrete update resides within the feasible region, PolyFlow fundamentally eliminates numerical integration overshoot. This allows for strict safety even with very few steps ($N=10$) while bypassing expensive iterative projection solvers.
>
> ## Q2. Ablation on the Constraint Encoding Architecture
>
> To clarify the role of the constraint encoding architecture, we conducted an ablation study across three tasks. Since the set of linear inequalities is inherently unordered, any effective encoding must be **permutation-invariant**. We compared three configurations:
> 1. **Cross-Attention:** Our default architecture, where constraint embeddings are fused into the latent space via cross-attention.
> 2. **MLP-based:** Constraints are processed by an MLP, aggregated via a sum operator (to ensure permutation-invariant), and concatenated with trajectory features.
> 3. **Without Embedding (w/o):** In this case, no explicit constraint information is provided. The network relies solely on state inputs.
>
> **Table A: Performance Comparison on Maze2d and HalfCheetah**
> |**Task**|**Architecture**|**MMD**|**W**|**KL**|**Acc**|**TotalTime(s)**|
> |-|-|-|-|-|-|-|
> |Maze2d|Attention|6.20e-6|0.041|0.134|6.15e-3|2.11|
> | |MLP| 2.62e-6| 0.033| 0.050| 6.37e-3 | 0.58|
> | |w/o|1.91e-6|0.033|0.066|6.03e-3|0.50|
> |HalfCheetah|Attention| 3.60e-2 | 0.857 | 2.643 | 2.708 | 0.089|
> | |MLP|2.98e-2|0.835|2.451|2.735|0.061|
> | |w/o|3.80e-2|0.836|2.471|2.717|0.060|
>
> **Table B: Rollout Performance on Unitree Go2 (Dynamic Constraints)**
> |**Architecture**|**Episode Length**|**Tracking Error**|
> |-|-|-|
> |Attention|305+-37|0.026+-0.007|
> |MLP|311+-64|0.026+-0.007|
> |w/o|227+-15|0.046+-0.008|
>
> **Key Insights**:
> - **Architectural Robustness**: For static tasks, the choice of constraint encoder (MLP vs. Attention) has a negligible impact on generative quality and safety. This proves PolyFlow is architecture-agnostic.
> - **The Critical Role of Encoding in Dynamic Tasks:** For tasks with static constraints (Maze2d, HalfCheetah), the "w/o" baseline performs surprisingly well. We hypothesize that the model implicitly learns the fixed boundaries from the state distribution. However, in the Go2 task, where constraints are state-dependent and time-varying, the performance gap widens significantly. Explicit encoding becomes essential for the model to generalize to shifting safe boundaries.
>
> **Core Contribution:** These results reinforce our primary claim: the performance of PolyFlow is rooted in the **discrete-time flow matching formulation** and the **projection-free ray-shooting operator**, rather than specific architectural tuning. We utilize the attention-based encoder in the manuscript to maintain maximum generality and demonstrate the framework's ability to handle highly complex, dynamic safety requirements.

---

> > ### Author Rebuttal · Reviewer_6UyV · 2026-04-03
> >
> > My concerns are well addressed, and I will maintain my positive rating

---

### Official Review · Reviewer_rQzw · 2026-03-15

**Soundness:** 3
**Presentation:** 3
**Significance:** 3
**Originality:** 3
**Overall Recommendation:** 5
**Confidence:** 4

**Summary:**

This paper presents PolyFlow, a flow-matching framework that strictly satisfies convex polytope constraints. The core idea is to enforce the learned flow map to remain exactly within a constructed convex feasible set through discrete flow matching and a ray-shooting-based parameterization. The authors claim that PolyFlow achieves zero constraint violations while maintaining computational efficiency in maze planning, safe control in MuJoCo, and quadruped locomotion under friction-cone constraints.

**Compliance With Llm Reviewing Policy:**

Affirmed.

**Final Justification:**

My concerns have been addressed.

**Key Questions For Authors:**

1. In Appendix F, the paper explicitly states that the maze dataset may contain trajectory points that violate the stricter constraints. If both vanilla flow and PolyFlow are trained on this dataset, then the comparison in the narrowed-corridor test environment described in Appendix F may be unfair to vanilla flow, since PolyFlow has access to the strictly constrained corridor information at inference time whereas vanilla flow does not. Could the authors provide a more detailed clarification on this point?
2. PolyFlow uses substantially fewer sampling steps than the other baselines, so its advantage in total runtime may largely stem from the smaller step count. Could the authors provide a comparison of the methods under the same number of sampling steps?

**Limitations:**

No. The paper should explicitly discuss the convex-polytope assumption, dependence on feasible initial states, and the gap between open-loop safety and closed-loop rollout safety.

**Strengths And Weaknesses:**

Strengths
1.The proposed ray-shooting plus scalar-gating parameterization is simple and appealing, and it imposes convex polytope constraints on the flow-matching framework in an elegant manner.
2.Combining discrete flow matching, strict safety constraints, and a projection-free parameterization is nontrivial, and the paper also provides theoretical derivations that appear to be sound, with no obvious gaps based on my inspection.

Weaknesses
see questions.

---

> ### Author Rebuttal · Authors · 2026-03-30
>
> We sincerely thank you for your insightful and constructive comments. Your feedback is incredibly valuable to us. Below, we address your questions in detail.
>
> ## Comparison Under the Same Number of Sampling Steps (Q2)
>
> You raised a great point regarding whether PolyFlow's runtime advantage stems primarily from utilizing fewer sampling steps. To investigate this, we conducted an additional Maze2D experiment restricting all baselines to **exactly 10 steps** (matching PolyFlow).
>
> **Table: Performance in Maze2D with identical sampling steps (step=10)**
>
> | **Metric**       | Flow    | FlowTrunc | SafeFlow | RoSD    | ReSD    | TVSD    | PolyFlow | PolyFlow-mlp |
> | - | - | - | - | - | - | - | - | - |
> | **Safety**       | 0.845   | 0         | 0.53     | 0.99    | 0.145   | 0       | **1**    | **1**        |
> | **MMD**          | 3.50e-5 | 2.71e-2   | 6.55e-3  | 1.76e-4 | 5.46e-2 | 9.01e-2 | 6.20e-6  | **2.62e-6**  |
> | **W**            | 5.41e-2 | 8.95e-1   | 3.48e-1  | 1.24e-1 | 1.8     | 2.45    | 4.08e-2  | **3.30e-2**  |
> | **KL**           | 2.66e-1 | 7.86      | 3.14     | 5.02e1  | 2.72    | 3.74    | 1.34e-1  | **5.04e-2**  |
> | **TotalTime(s)** | 0.324   | 0.363     | 7.558    | 1.12    | 8.926   | 2.558   | 2.11     | 0.582        |
>
>
> **Analysis of Results:**
>
> - **Quality & Safety:** When restricted to 10 steps, the generation quality of baselines degrades severely, and they fail to achieve strict safety (often trapped in local optima due to conflicts between denoising dynamics and constraint projections). PolyFlow, however, perfectly maintains safety and vastly outperforms the others in trajectory quality.
>
> - **Why we didn't fix steps originally:** We initially reported baselines using the step counts recommended in their respective papers to ensure a fair representation of their best generation quality. As shown above, forcing them to 10 steps ruins their performance.
>
> - **Inference Efficiency:** While slightly slower than naive FlowTrunc, PolyFlow's time complexity scales linearly with constraint count (via ray-shooting), avoiding the polynomial time explosion typical of QP solvers.
>
> - **Further Acceleration:** To demonstrate architectural  flexibility, we replaced our Cross-Attention constraint embedding with a lightweight MLP (`PolyFlow-mlp`). This variant achieved similar high-quality, strictly safe results while reducing inference time to 0.582s, proving PolyFlow can be highly optimized for real-time applications.
>
>
> ## Fair Comparison with Vanilla Flow on the Maze Dataset (Q1)
>
> You correctly noted that the original dataset contains trajectories violating the stricter narrowed-corridor constraints, which might seem unfair to Vanilla Flow. We would like to clarify that Vanilla Flow was included purely as a **reference for unconstrained generation quality**, rather than as a baseline for constraint satisfaction. It is expected to violate constraints.
>
> To prove that explicitly constraint embedding is fundamentally necessary—even when trained on perfectly clean data—we conducted a new experiment. We filtered the training dataset to remove all unsafe trajectories (retaining only the strictly safe 67%) and trained a new unconstrained model, **FlowFiltered**.
>
> **Table: Vanilla Flow vs. FlowFiltered (step=10)**
>
> | **Metric** | Flow(step=10) | FlowFiltered(step=10) | PolyFlow (step=10) |
> | ---------- | ------------- | --------------------- | ------------------ |
> | **Safety** | 0.845         | 0.965                 | **1.0**            |
> | **MMD**    | 3.50e-5       | 4.33e-5               | **6.20e-6**        |
> | **W**      | 5.41e-2       | 5.55e-2               | **4.08e-2**        |
> | **KL**     | 2.66e-1       | 2.30e-1               | **1.34e-1**        |
>
> Even when trained exclusively on safe data, `FlowFiltered` fails to achieve strict safety (Safety = 0.965) due to inherent fitting and discretization errors. This confirms that data filtering alone is insufficient, underscoring the necessity of PolyFlow's explicit constraint embedding.
>
> ## Limitations
>
> We agree with your suggestion and will expand our Limitations section to explicitly discuss:
>
> 1. **Convex-Polytope Assumption:** PolyFlow currently requires constraints to be formulated as convex polytopes.
> 2. **Initial States:** Guaranteeing strict satisfaction requires the initial distribution to fall within the feasible region, which can be handled by uniform sampling within the constraint polytope's Chebyshev ball.
> 3. **Open-Loop vs. Closed-Loop Safety Gap:** If constraints are defined on the action space (e.g., the HalfCheetah), PolyFlow guarantees zero violations during execution. However, when constraints are defined on the state space, a gap emerges during closed-loop execution. This is due to the inherent mismatch between the learned empirical dynamics and the true environmental dynamics—a broader challenge in generative models expected to mitigate with larger datasets.
>
> We sincerely hope these answers address your concerns.

---

> > ### Author Rebuttal · Reviewer_rQzw · 2026-04-04
> >
> > Thank you for addressing my concerns. I will raise my score.

---

> > > ### Author Response · Authors · 2026-04-05
> > >
> > > We sincerely thank the reviewer for the positive feedback and for the time spent evaluating our rebuttal.

---

### Decision · Program_Chairs · 2026-04-30

**Decision:**

Accept (regular)

**Comment:**

This paper proposes PolyFlow, a discrete-time flow matching framework that guarantees strict satisfaction of polyhedral constraints via a projection-free ray-shooting parameterization.

Reviewers unanimously recognized the mathematical formulation. The implicit construction of a safe vector field is conceptually novel. Furthermore, the empirical evaluation is robust, featuring both standard benchmarks and real-robot experiments (Unitree Go2) that convincingly demonstrate the framework's practical significance and efficiency.

The authors provided a strong rebuttal that fully resolved all initial questions, leading all four reviewers to explicitly confirm their satisfaction (with several raising their scores). Key clarifications included:
- The authors demonstrated that continuous-time ODE solvers (e.g., RK4) inevitably suffer from numerical integration overshoot near boundaries, validating the necessity of their discrete-time approach for strict safety.
- Additional experiments showed that PolyFlow vastly outperforms baselines even when strictly restricted to identical sampling steps. They also showed that data filtering alone fails to guarantee safety, proving the necessity of explicit constraint embedding.
- The authors successfully clarified the step-size/direction coupling in their architecture, provided theoretical extensions for equality constraints, and thoroughly justified the real-world applicability of linear safety polytopes in complex robotic tasks.

Therefore, the paper is recommended for acceptance. Please incorporate the suggestions and additional experiments and clarifications into the camera-ready version.